# Mixed-Lineage Leukemia 1 Inhibition Enhances the Differentiation Potential of Bovine Embryonic Stem Cells by Increasing H3K4 Mono-Methylation at Active Promoters

**DOI:** 10.3390/ijms241511901

**Published:** 2023-07-25

**Authors:** Chen Li, Xuejie Han, Jing Wang, Fang Liu, Yuanyuan Zhang, Zihong Li, Zhenyu Lu, Yongli Yue, Jinzhu Xiang, Xueling Li

**Affiliations:** State Key Laboratory of Reproductive Regulation and Breeding of Grassland Livestocks, Inner Mongolia University, Hohhot 010070, China; 21908024@mail.imu.edu.cn (C.L.); hxj3876@126.com (X.H.); nnlrl@mail.imu.edu.cn (J.W.); 21908023@mail.imu.edu.cn (F.L.); m15024727853_2@163.com (Y.Z.); lornee_lee@sina.com (Z.L.); luzhenyu0312@163.com (Z.L.); yueyongli228@163.com (Y.Y.); xiangjz0214@sina.com (J.X.)

**Keywords:** bovine embryonic stem cells (bESCs), pluripotent, H3K4me1 modification, DNA methylation modification

## Abstract

Mixed-lineage leukemia 1 (MLL1) introduces 1-, 2- and 3-methylation into histone H3K4 through the evolutionarily conserved set domain. In this study, bovine embryonic stem cells (bESCs, known as bESCs-F7) were established from in vitro-fertilized (IVF) embryos via Wnt signaling inhibition; however, their contribution to the endoderm in vivo is limited. To improve the quality of bESCs, MM-102, an inhibitor of MLL1, was applied to the culture. The results showed that MLL1 inhibition along with GSK3 and MAP2K inhibition (3i) at the embryonic stage did not affect bESCs’ establishment and pluripotency. MLL1 inhibition improved the pluripotency and differentiation potential of bESCs via the up-regulation of stem cell signaling pathways such as PI3K-Akt and WNT. MLL1 inhibition decreased H3K4me1 modification at the promoters and altered the distribution of DNA methylation in bESCs. In summary, MLL1 inhibition gives bESCs better pluripotency, and its application may provide high-quality pluripotent stem cells for domestic animals.

## 1. Introduction

Mixed-lineage leukemia 1 (MLL1), also known as MLL, KMT2A, HRX, HTRX and ALL1, is one of the six mixed-lineage leukemia (MLL) family histone methyltransferases (HMT) in mammals [1,2,3]. It mainly introduces 1-, 2- and 3-methylation into histone H3K4 through the evolutionarily conserved set domain. MLL1 and H3K4 methylation (H3K4me) take place in the promoter, transcription initiation site and 5′ transcription region of target genes and promote transcription initiation, so they play an important role in transcriptional regulation, especially in the early development of zygotic gene activation (ZGA) and hematopoiesis [4,5].

The H3K4 HMT activity of MLL1 is controlled by the core complex, which is composed of MLL1 and WDR5. The activity of MLL1 alone is weak, but its H3K4 HMT activity is greatly enhanced with the formation of the core complex [6]. The MLL1 core complex consists of WDR5 and MLL1 proteins [6,7]. By disrupting the protein interaction between WDR5 and MLL1, the MLL1 core complex can be effectively dissociated and MLL1 activity can be inhibited [8]. MM-102 is one of the compounds that can prevent the interaction between MLL1 and WDR5, inhibiting MLL1 activity. When mouse bone marrow cells transfected with the *MLL1-AF9* fusion gene were cultured with MM-102, the expression of *MLL1* target genes *HoxA9* and *Meis-1* was greatly reduced [9]. Furthermore, down-regulating H3K4me3 through MM-102 can improve ZGA and abnormal expression patterns of epigenetic chromatin-modifying enzymes, as well as pluripotency and apoptosis genes in the blastocyst stage, and greatly improve efficiency and embryo quality in porcine somatic cell nuclear transfer, making it closer to that of in in vivo embryos [10]. Our latest research found that adding appropriate concentrations of MM-102 and 3i (MM-102, PD0325901 and CHIR99021) during the ZGA stage of mouse and bovine in vitro fertilization (IVF) embryo development can greatly improve the development of IVF blastocysts without affecting their quality [11]. MM-401 is another inhibitor of MLL1, and can transform mouse EpiSCs into a naïve pluripotent state [12]. MLL1 inhibition causes the redistribution of H3K4me1 in enhancers, germline determinants and EpiSC markers to regulate the pluripotency regulatory network and restore the ability of EpiSCs to participate in embryonic development and germline chimerism [12]. However, there is still a lack of relevant experimental studies on whether MLL1 inhibition by MM-102 alone or combined with other small-molecule inhibitors can transform human or livestock prime ESCs into a naïve state.

Cattle provide humans with high-quality meat products and nutritious dairy products, which have important economic and research value. The establishment of bovine pluripotent stem cells (bPSCs) has great significance for germplasm conservation, gene editing, breeding and understanding the developmental specificity of ruminants. In 2018, Bogliotti et al. obtained bovine prime ESCs, known as CTFR-bESCs, in a culture system supplemented with fibroblast growth factor 2 (FGF2) and WNT signaling pathway inhibitor IWR1, which exhibited characteristics of prime PSCs [13]. Recently, bovine expanded pluripotent stem cells were established from blastocysts or via the reprogramming of bovine embryonic fibroblasts, which possess embryonic and extraembryonic development potential [14]. However, whether we can transform bovine prime ESCs into a naïve state remains undetermined.

In this study, bovine prime ESCs were established from normal cultured IVF embryos, or those treated with MM-102 or 3i (MM-102, PD0325901 and CHIR99021)-. MM-102 improved the pluripotency and enhanced the capacity of the differentiation of bESCs. At the same time, after the addition of MM-102, bESCs could differentiate the endoderm in teratoma in vivo, and the signaling pathways of pluripotency were activated by MM-102 treatment. The effects of MLL1 inhibition on H3K4 methylation were determined via RNA-sequencing, Western blotting and ChIP-Seq analysis of H3K4me1. The modification pattern of H3K4me1 was altered by MM-102. In particular, the quantity of H3K4me1 in the promoter region of the genes was decreased, but the total H3K4me1 modification was significantly increased in the treated cells. Moreover, DNA methylation was comprehensively investigated via genome-wide DNA methylation sequencing. The pattern of DNA methylation modification was also significantly altered after MM-102 treatment, the methylation level of the promoter was reduced, and the expression of DNMT3B was significantly increased. This study will provide new insight into stem cell state transformation through the regulation of epigenetic modification.

## 2. Results

### 2.1. MLL1 Combined with Gsk3 and Map2k Inhibition (3i) at Embryonic Stage Did Not Affect bESCs’ Establishment and Pluripotency

Previously, we found that MLL1 combined with GSK3 and MAP2K inhibition improves the development and the quality of in vitro-fertilized embryos [11]. In this study, a CTFR-bESC culture system was applied to establish bovine- ESCs from MM-102 (50 μM)-, 2i (0.5 μM PD0325901 and 3 μM CHIR99021)- and 3i (2i plus 30 μM MM-102)-treated bovine IVF embryos (Figure 1A). When embryos formed blastocysts at 7–8 days, the blastocysts, removed from the zona pellucida, were plated to establish bESCs using mouse fetal fibroblast (MEF) feeder cells. The obtained bESC cell lines were named bESCs-102, bESCs-2i and bESCs-3i (Figure 1A and Appendix A). The establishment rate of bESCs-F7 by CTFR was 6.82%, while the establishment rates of bESCs-2i, bESCs-102 and bESCs-3i were increased to 14.29%, 10.00% and 8.33%, respectively (Appendix A). Although the IVF embryos that inhibited GSK3 and MAP2K generated more bESCs cell lines, the bESCs-2i cells proliferated slowly with weak alkaline phosphatase (AP) staining, and they hardly survived subculture (Appendix A). bESCs-102, where MLL1 was inhibited, also grew slowly and showed weak AP staining, but this cell line could form colonies with clear edges and survived subculture for more than 20 passages (Figure 1B). bESCs-3i, with simultaneously inhibited MLL1, GSK3 and MAP2K, was similar to bESCs-F7, exhibiting strong AP-positive cells and forming colonies with clear edges, and survived in subculture for more than 60 passages (Figure 1B). All of the cell lines maintained a normal karyotype, with 60 chromosomes (Figure 1B). Immunofluorescence staining (IF) analysis revealed that bESCs-F7, bESCs-102 and bESCs-3i expressed the pluripotency transcription factors OCT4, SOX2, NANOG and SSEA4 (Figure 1C). For SSEA1, which was considered a naïve pluripotency marker, only a few positive cells could be observed in bESCs-F7, and more fluorescent cells could be found in bESCs-102 and bESCs-3i (Figure 1C). At the same time, regarding TRA-1–60, a prime pluripotency marker, positive cells could be weakly observed in bESCs-F7 and bESCs-3i, and regarding TRA-1-81, a naïve pluripotency marker, positivity was strongly observed only in bESCs-102 (Appendix A). To confirm the change in pluripotency, the proportion of SSEA4- and SSEA1-positive cells in bESCs was analyzed via flow cytometry. The results showed that bESCs-F7 contained 98.12% SSEA4-positive cells, while the percentages in bESCs-102 and bESCs-3i were reduced to 97.23% and 85.21% (Figure 2A). bESCs-F7 contained 58.90% SSEA1-positive cells, and the percentages of bESCs-102 and bESCs-3i were reduced to 18.53% and 40.24% (Figure 2A). We performed transcriptome analyses on bESCs-F7, bESCs-102 and bESCs-3i via RNA sequencing (RNA-seq). The analysis of naïve and prime genes revealed that bESCs-F7 and bESCs-3i showed similar pluripotent gene expression patterns, and bESCs-102 showed higher expression levels in naïve and prime marker genes (Figure 2B). The expression of pluripotency genes such as *OCT4*, *SOX2*, *NANOG*, *NCAM1*, *TET1*, *STELLA*, *REX1* and *TEAD4* was also confirmed via quantitative real-time PCR (qRT-PCR), and no significant difference between bESCs-F7 and bESCs-3i was found (Figure 2C). However, the expression of these genes was slightly lower in bESCs-102 cells, and the expression level of *REX1* was higher in bESCs-3i (Figure 2C).

To detect the differentiation ability of the cells, in vitro embryoid body (EB) formation and in vivo teratoma formation experiments were performed. The results showed that bESCs-102 and bESCs-3i cells differentiated into three germ layer cells in vitro, similarly to bESCs-F7 (Appendix A). However, bESCs-102 and bESCs-3i cells easily formed teratomas when injected into NOD-SCID mice in vivo, and the teratoma percentage was significantly higher for bESCs-102 (46.15%) and bESCs-3i (92.86%) than for bESCs-F7 (13.33%) (Appendix A). Furthermore, the size of the teratoma formed from bESCs-3i was bigger, and the derivatives of the ectoderm and mesoderm could be easily detected (Figure 2D). bESCs-102, bESCs-2i and bESCs-3i have almost no endodermal derivatives in teratoma formation in vivo.

In summary, MLL1 inhibition, and MLL1 combined with GSK3 and MAP2K inhibition (3i), did not affect bESCs’ establishment or pluripotency, and bESCs established from 3i-treated bovine embryos had better differentiation potential. In other words, bESCs-102 showed a greater ability to express pluripotency genes.

### 2.2. MLL1 Inhibition Improved the Differentiation Potential of CTFR-bESCs

Given that MM-102 treatment of embryos altered the pluripotency of the established bESCs, we wondered whether MM-102 could induce similar changes in the bESCs-F7 culture system. Different concentrations of MM-102 (0 μM, 10 μM, 30 μM, 50 μM and 70 μM) were added to CTFR, and the cells exhibited slow proliferation, loose colonies and massive cell death when MM-102 was added in quantities over 10 μM. After screening, we confirmed two protocols; one involved the addition of 5 μM of MM-102 to CTFR in long-term cultures, and in the other, bESCs were cultured for 7 days with the addition of 50 μM MM-102, and then switched back to CTFR. The derived cell lines were named bESCs-102-5 and bESCs-102-50 (Figure 3A). bESCs-102-5 and bESCs-102-50 had an AP-positive and correct karyotype, and clones of bESCs-102-5 had clearer edges (Figure 3B). The protein expression of OCT4, SOX2, NANOG, SSEA1 and SSEA4 were determined via IF (Figure 3C). TRA-1-60 could be detected in both bESCs-102-5 and bESCs-102-50, and TRA-1-81 was observed in bESCs-102-5 but not in bESCs-102-50 (Appendix A). We also examined the SSEA4- and SSEA1-positive cells in bESCs-102-5 and bESCs-102-50, via flow cytometry. The proportion of SSEA4 increased from 67.72% (bESCs-F7) to 98.93% (bESCs-102-5) and 93.11% (bESCs-102-50) (Figure 4A). The proportion of SSEA1 in bESCs-F7 was 4.20%, and it increased to 65.69% in bESCs-102-5 and 39.90% in bESCs-102-50 (Figure 4A). To further investigate the pluripotency of MM-102-treated cells, we analyzed the expression of typical naïve and prime pluripotent markers identified in human ESCs after transcriptome sequencing. We found that most of the prime pluripotency markers and some of the naïve markers were upregulated in bESCs-102-5 and bESCs-102-50 (Figure 4B). The qRT-PCR results showed that *OCT4* and *NANOG* were significantly up-regulated after MM-102 was added. However, the expression of *SOX2* was decreased in bESCs-102-5 and increased in bESCs-102-50. The prime gene *FGF4* was up-regulated in bESCs-102-5 and bESCs-102-50, while other prime genes, like *NCAM1* and *C-MYC*, showed no significant differences or down-regulation. Meanwhile, the expression of naïve genes, such as *TFCP2L1*, *REX1*, *STELLA* and *HOXA9*, was significantly increased (Figure 4C).

We examined the differentiation ability of bESCs-102-5 and bESCs-102-50 in vitro and in vivo. bESCs-102-5 and bESCs-102-50 exhibited spontaneous differentiation ability in the formation of EBs, and continued to differentiate the ectoderm (GFAP), mesoderm (SMA) and endoderm (AFP) in vitro (Appendix A). The teratoma formation percentage of bESCs-102-5 (26.32%) and bESCs-102-50 (29.41%) was higher than that of bESCs-F7 (13.33%) (Appendix A). The teratomas formed from bESCs-102-5 and bESCs-102-50 were bigger than those from bESCs-F7, and endoderm derivatives were detected in bESCs-102-5 (Figure 4D).

The ability of MM-102 to improve bESC pluripotency was stronger in ESCs than in embryos. In addition, it improved prime and naïve pluripotency in a more comprehensive way, rather than focusing on a certain pluripotency state. The inhibition of MLL1 significantly enhanced the differentiation ability of bESCs.

### 2.3. Inhibition of MLL1 Up-Regulated the Expression of Genes Involved in Stem Cell-Related Signaling Pathways

To determine whether MLL1 inhibition affects the pluripotency of bESCs, we thoroughly investigated the RNA-seq data. Regarding bESCs obtained from MM-102-treated embryos, principal component analysis (PCA) showed that bESCs-F7 and bESCs-3i were grouped together according to their PC2 values, and bESCs-102 was slightly different from them (Figure 5A). Among the 2063 differentially expressed genes, 785 genes were up-regulated and 1278 genes were down-regulated in bESCs-102 compared with bESCs-F7 (Figure 5B, left). We enriched the up-regulated genes of bESCs-102 using KEGG analysis and found that they were mainly concentrated in the classical signaling pathways that regulate the pluripotency of stem cells, such as the WNT and TGF-beta signaling pathways (Figure 5C, top). An analysis of the bioprocesses of GO analyzed showed that up-regulated genes in bESCs-102 were mainly involved in the biological processes related to axon development neurogenesis and the neuron development of bESCs-102 (Appendix A), and down-regulated genes in bESCs-102 inhibited the morphogenesis of mesoderm- and endoderm-related structures, such as blood vessels, the skeletal system, collagen fibril organization and animal organ morphogenesis (Appendix A). However, in bESCs-3i, only 287 genes were up-regulated and 844 were down-regulated compared with bESCs-F7 (Figure 5B, right). Most of the genes whose expression increased in bESCs-3i participated mainly in glycolysis/gluconeogenesis, the biosynthesis of amino acids/carbon metabolism and other metabolism-related signaling pathways (Figure 5C, middle). Genes whose expression decreased in bESCs-3i were more involved in anatomical structure morphogenesis and regulation of the developmental process (Appendix A). Interestingly, genes that were highly expressed in bESCs-F7 were concentrated in the PI3K-AKT signaling pathway (Figure 5C, bottom).

The PCA of bESCs-102-5, bESCs-102-50 and bESCs-F7 based on RNA-seq data showed that they were similar, but the data repeatability of bESCs-102-5 and bESCs-102-50 was better than that of bESCs-F7; this shows that MM-102 confers better homogeneity to bESCs (Figure 6A). The gene expression difference was examined via scatterplots, which show that 1194 genes were down-regulated and 656 genes were up-regulated in bESCs-102-5 (Figure 6B, left). They also show that 818 genes were down-regulated and 460 genes were up-regulated in bESCs-102-50 (Figure 6B, right). GO analysis of the biological process showed that the down-regulated genes in bESCs-102-5 were mainly involved in developmental process regulation, including sensory development, sensory organ development, and the development of extracellular structure organization (Appendix A). KEGG analyses were performed on the highly expressed genes in bESCs-F7, bESCs-102-5 and bESCs-102-50, respectively, and 234 genes in bESCs-F7 were associated with oxidative phosphorylation, inflammatory processes, RNA damage, and DNA damage (Figure 6C, top). MAPK and PI3K-Akt signaling pathway-related genes, such as *TEK*, *TGFA*, *MAPK10*, *FGF8* and others, were highly expressed in bESCs-102-5 (Figure 6C, middle). The specificity of bESCs-102-50 is reflected in the WNT and mTOR signaling pathways (Figure 6B, bottom).

MLL1 inhibition improves pluripotency via up-regulation of the classical pluripotent stem cell signaling pathways. The WNT signaling pathway was up-regulated in both bESCs-102 and bESCs-102-50. However, the MAPK and PI3K-AKT signaling pathways were up-regulated in bESCs-102-5, which had the strongest pluripotency.

### 2.4. Inhibition of MLL1 Increased the Proportion of H3K4me1 in the Promoter Region of bESCs

To reveal the effects of MM-102 on H3K4 modification in bESCs, the expression of related enzymes was analyzed. The expression of MLL1 was reduced by half after 50 μM MM-102 was added into the cell culture system, but 5 μM MM-102 had no significant effect on MLL1 expression (Figure 7A). However, the expression of MLL1 was almost completely inhibited by the addition of MM-102 to the embryo, but its expression was significantly increased after being combined with GSK3 and MAP2K inhibitors (3i) (Figure 7A). Meanwhile, the histone lysine methyltransferase PRDM14 was up-regulated to varying degrees by MM-102, with greater variation in the bESCs-102-5 and bESCs-102-50 groups than in the bESCs-102 and bESCs-3i groups (Figure 7B). The Western blotting results showed that the protein levels of PRDM14 were increased in bESCs-102-5 and decreased in bESCs-102, compared with bESCs-F7 (Figure 7C). The protein levels of G9A, which is one of the histone methylation transferases, were lower in bESCs-102-50 and higher in bESCs-102 (Figure 7C). The protein level of histone acetyltransferase P300 was increased in bESCs-102-5 and bESCs-102-50, but decreased in bESCs-102 and bESCs-3i (Figure 7C). Methylation-related genes were also analyzed using transcriptome sequencing data, and these genes showed higher expression in all three MM-102 treatment groups and bESCs-3i than in bESCs-F7 (Appendix A). Compared with bESCs-F7, the genes of ubiquitination- and acetylation-related enzymes were more active in bESCs-102, bESCs-102-5 and bESCs-102-50, while they showed the same activity level in bESCs-3i as in bESCs-F7 (Appendix A). We further explored the effects of MLL1 inhibition in bESCs on global H3K4me1 and H3K4me2 via Western blotting. H3K4me1 increased in bESCs-102-5 and bESCs-102-50 and decreased significantly in bESCs-102 and bESCs-3i. H3K4me2 significantly increased in bESCs-102-5 and slightly increased in bESCs-102 and bESCs-3i (Figure 7D). However, H3K27me3 increased in all MM-102-treated cell lines (Figure 7D). It can be concluded that treatment with MM-102 at different stages of the establishment of bESCs has different influences on the methylation modification of bESCs.

We then focused on H3K4me1 modification changes in bESCs-102-5 and bESCs-102-50 via ChIP-seq analysis. The PCA results showed that H3K4me1 modification in bESCs and bovine fetal fibroblasts (BFFs) was completely different, and the addition of MM-102 changed the distributions of H3K4me1 modification in bESCs-F7 (Figure 7E). Compared with BFFs, the total amount of H3K4me1 in bESCs was lower (Figure 7F). After treatment with MM-102, the overall H3K4me1 level in bESCs-102-5 and bESCs-102-50 was decreased and more concentrated at the transcription start site (TSS) (Figure 7F). Compared with bESCs-102-5, the H3K4me1 of bESCs-102-50 was more enriched in TSS, and lower in other gene regulatory regions (Figure 7F). GO analysis showed that the biological processes of genes enriched with H3K4me1 in bESCs-102-5 compared with bESCs-F7 included the negative regulation of the macromolecule biosynthetic process, the negative regulation of gene expression, and the negative regulation of transcription and embryonic organ development (Appendix A). Compared with bESCs-102-5 and bESCs-F7, the biological processes of genes enriched with H3K4me1 in bESCs-102-50 include the regulation of muscle system processes, neurogenesis, cell–cell signaling, tube development, nervous system development and cellular response stimuli (Appendix A). Compared with bESCs-102-5 and bESCs-102-50, the biological processes of genes enriched with H3K4me1 in bESCs-F7 included circulatory system development, intracellular signal transduction, the anatomical structure formation involved in morphogenesis and the regulation of animal organ morphogenesis (Appendix A). KEGG analysis showed that compared with bESCs-F7 and bESCs-102-5, the related signaling pathways of genes enriched with H3K4me1 in bESCs-102-50, including animal autophagy, lysosome and protein processing in the endoplasmic reticulum and sphingolipid signaling pathway (Appendix A). However, there were no significant differences between bESCs-102-5 and bESCs-F7 after KEGG analysis (Appendix A).

We further analyzed the H3K4me1 modification patterns in BFFs, bESCs-F7, bESCs-102-5 and bESCs-102-50. bESCs-102-5 showed similar H3K4me1 modification to bESCs-F7 in the 10-100kb region upstream and downstream of the TSS, and H3K4me1 modification in bESCs-102-50 was similar to that in BFFs (Appendix A). The distribution of H3K4me1 in each gene regulatory region was also analyzed (Figure 8A). The percentage of H3K4me1 in the promoter region was slightly increased in bESCs-102-50 compared with bESCs-F7, but it was significantly increased in bESCs-102-5 (Figure 8A). Although the distribution of H3K4me1 at the promoters was increased by MLL1 inhibition, the total amount of H3K4me1 enriched at the promoter position decreased, and bESCs-102-50 was more obvious than bESCs-102-5 (Figure 8B). A volcanic map was drawn for the genes enriched with H3K4me1 in the promoter region, and the difference between bESCs-102-50 and bESCs-F7 (1469 different genes) was greater than that between bESCs-102-5 and bESCs-F7 (603 different genes) (Figure 8C). Among these, compared with bESCs-F7, 327 genes showed increased enrichment and 276 genes were down-regulated in bESCs-102-5-, while 729 genes showed increased enrichment and 740 genes were down-regulated in bESCs-102-50 -(Figure 8C). Analyses of the peak patterns of the promoter regions of *OCT4*, *DUSP5*, *KLF4* and *PRDM14* showed a decrease in the number of peaks clustered in bESCs-102-5 and bESCs-102-50 (Figure 8D and Appendix A). Enrichment with H3K4me1 in the whole genes of *NANOG*, *OCT4*, *KLF4* and *DUSP5* was increased in bESCs-102-5 and bESCs-102-50 and showed obvious differences in distribution (Figure 8E). Heat maps of the related enriched naïve and primed genes showed high similarity between bESCs-F7 and bESCs-102-50 (Appendix A). In bESCs-102-5, H3K4me1 modification of the naïve genes *CD9*, *FGF4* and *DPPA3* was increased (Appendix A), and H3K4me1 modification of the primed genes *DUSP6*, *ZIC2*, *C-MYC* and *HOXB3* was also increased (Appendix A). In summary, MLL1 inhibition decreased and changed the pattern of H3K4me1 modification in bESCs, which may lead to improved pluripotency in bESCs.

### 2.5. MLL1 Inhibition Down-Regulated mCG Levels in the Promoter Region of bESCs

To further investigate the influences of the MLL1 inhibition on bESCs, we analyzed the alteration of DNA methylation in MM-102-treated cells. At first, the expression of DNMT family genes related to DNA methylation was detected via qRT-PCR. After MLL1 inhibition, the level of *DNMT3A* was down-regulated, but that of *DNMT3B* was up-regulated. Interestingly, *DNMT1* and *DNMT3L* in bESCs-102-5 and bESCs-102-50 were down-regulated, while bESCs-102 and bESCs-3i were up-regulated (Figure 9A). The protein of DNMT3A was down-regulated and that of DNMT3B was up-regulated in bESCs-102-5 and bESCs-102-50. In bESCs-102, both DNMT3A and DNMT3B were up-regulated, whereas in bESCs-3i, both were down-regulated (Figure 9B).

Next, whole-genome methylation sequencing (WGMS) of the cells was performed. The results showed that the methylated regions of bESCs-102-5 and bESCs-102-50 are more widely distributed than those of bESCs-102 and bESCs-3i (Figure 9C). Moreover, at the methylation level, bESCs-102 and bESCs-3i were more strongly correlated with bESCs-F7, while bESCs-102-5 and bESCs-102-50 were considerably different from bESCs-F7 (Figure 9C,D). The methylation regions of bESCs-102-5 and bESCs-102-50 were more concentrated than those of bESCs-F7. bESCs-102 and bESCs-3i had a higher proportion of less-methylated regions than bESCs-F7 (Appendix A). DMR analysis of the methylation of CpG showed that MLL1 inhibition changed the methylation distribution of bESCs, and, compared with bESCs-F7, all the cell lines treated with MM-102 showed different patterns of CpG DMR methylation; however, the pattern of bESCs-102-5 was similar to that of bESCs-102-50, and the pattern of bESCs-102 was similar to that of bESCs-3i (Appendix A). Heat maps showed that MLL1 inhibition induced higher levels of methylation in the CpG region than bESCs-F7 (Appendix A). Notably, MM-102 decreased mCG levels in the upstream regulatory region and downstream regulatory region of bESCs. On the gene body, the degree of methylation of bESCs-102-50 and bESCs-102 was almost the same as that of bESCs-F7, while bESCs-102-5 methylation was higher than that of bESCs-F7, and bESCs-3i methylation was lower than that of bESCs-F7 (Appendix A). We further analyzed changes in the DNA methylation levels in each gene region. The methylation levels of the CpG island (CGI), promoter and exon in bESCs-102-5 and bESCs-102-50 were decreased compared with bESCs-F7, but the methylation levels of the CGI shore, intron and repeat regions was increased (Figure 9E). Regarding bESCs-102 and bESCs-3i, the methylation level of the promoter was also decreased, the difference being that, except for the promoter, the methylation level of bESCs-102 did not change, but the exon, intron, 3′-untranslated region (3′-UTR, utr3) and repeat regions of bESCs-3i decreased (Figure 9E). mCG in the promoter region of OCT4 and NANOG decreased upon MLL1 inhibition following bisulfite genomic sequencing of the promoter regions (Figure 9F). In short, the inhibition of MLL1 down-regulated the expression level of *DNMT3A*, up-regulated the expression level of *DNMT3B* and changed the distribution of DNA methylation modification; this was mainly reflected in the increase in the overall CpG methylation level, the decrease in mCG in the upstream and downstream regions of the genes and the decrease in mCG in the promoter region. These changes may partly explain why MM-102 changes the pluripotency of bESCs.

## 3. Discussion

Given the highly accessible and hyperactive chromatin structures in pluripotent stem cells (PSCs), it is generally assumed that H3K4me plays an important ‘‘housekeeping’’ role in PSCs and is necessary for PSCs to maintain self-renewal and unlimited differentiation potential [15]. Zhang et al. found that H3K4me1 was significantly different between mouse ESCs and EpiSCs, but H3K4me3 was not. Interestingly, the inhibition of MLL1 led to a genome-wide change in H3K4me1 in EpiSCs and the global redistribution of H3K4me1 at the enhancers, and repressed lineage determinant factors and EpiSC markers, which indirectly regulate the transcription of mouse ESCs [12]. Naïve PSCs are distinguished from primed PSCs by their capacity to form teratomas and chimeric animals following their introduction into pre-implantation embryos, and exhibit unique transcription makers and DNA methylation levels [15]. In recent years, significant advances have been made in the conversion of the primed state of hPSCs into naïve pluripotency. Human naïve pluripotency has shown better differentiation than the primed state, which is associated with the accumulation of DNA methylation and epigenetic-repressive marks in the primed state [16]. Previous studies showed that MLL1 inhibition transformed mouse EpiSCs into a naïve pluripotent state [12], and the combined treatment of the MLL1 inhibitor and 1α,25-dihydroxyvitamin D3 (1,25-(OH)2D3) enhanced the functionality of expanded PSCs, triggering an extended 2C-like state in vitro and robust totipotent-like properties in vivo [17]; this indicated the MLL1 inhibition increases the differentiation capacity of PSCs. In this study, we found that MLL1 inhibition improved the pluripotency and differentiation potential of bESCs by decreasing H3K4me1 modification at the promoters and altered the distribution of DNA methylation in bESCs. However, MLL1 inhibition increased some naïve and primed markers simultaneously, but did not transform bESCs from a primed to a naïve state as in the mouse EpiSCs. We also found that MLL1 inhibition by itself cannot as effectively transform bovine ESCs to a naïve state. This may be due to the following: firstly, the MLL1 inhibitor used in our study is different from that used at the Dou Lab [12], which may lead to differences in some effects; secondly, MM-102 only up-regulated some of the stem cell signaling pathways, such as PI3K-Akt and WNT, which may not be enough to transform bESCs from a primed to a naïve state; finally, the characteristics of PSCs from domestic animals are quite different from those from mice and humans, and further investigation is needed to thoroughly define the PSCs of domestic animals.

MLL1 epigenetically regulates gene expression patterns that specify cellular identity in both embryonic development and adult stem cell populations. *Mll1* is required for the expression of neurogenic—but not gliogenic—transcriptional modules in multipotent neural stem cell (NSC) populations, which further indicates that specific *Mll1*-dependent genes may be useful for direct reprogramming strategies [18]. MLL1 induced the activation of a Rac/Rho/Integrin signaling axis, to enhance hematopoiesis from murine embryonic stem cells [19]. In this study, MLL1 inhibition in bESCs increased the expression of PRDM14, which is a histone lysine methyltransferase and a common marker of primordial germ cells and pluripotent embryonic stem cells [20]. Moreover, the inhibition of MLL1 down-regulated the expression level of *DNMT3A* and up-regulated the expression level of *DNMT3B*. DNMT3B is the primary driver of de novo DNA methylation in actively transcribed genes, while DNMT3A plays a minimal role in ESCs [21]. These results indicate the effects of MLL1 inhibition in bESCs, mainly on an epigenetic level. MLL1 inhibition changed the distribution of DNA methylation, which was mainly reflected in the increase in the overall CpG methylation level, the decrease in mCG in the upstream and downstream regions of the genes, and the decrease in mCG in the promoter region. The increased expression of DNMT3B in bESCs is associated with increased DNA methylation (CpG). Meanwhile, mCG in the promoter region of pluripotent genes such as *OCT4 and NANOG* was decreased upon MLL1 inhibition, following the bisulfite genomic sequencing of the promoter regions. These changes may be one of the reasons why MM-102 changes the pluripotency of bESCs. Although the distribution of DNA methylation was changed following MLL1 inhibition, the global DNA hypomethylation observed in naïve PSCs was not achieved. To obtain real naïve PSCs for bovines, a thorough understanding of pluripotency control in ruminants is needed.

In conclusion, MLL1 inhibition enhanced the pluripotency of bovine embryonic stem cells, which enriched the promoter of pluripotency genes with H3K4me1 to increase the expression of pluripotency genes, and MM-102 altered the global DNA methylation distribution of bESCs.

## 4. Materials and Methods

### 4.1. Animal Care and Use

All experiments with mice (generation of embryonic fibroblasts and teratoma formation) were conducted in accordance with the Guide for Care and Use of Laboratory Research Involving Animals and approved by the Inner Mongolia University’s Animal Care and Use Committee (approval code: IMU-mouse-2021-037, approval date: 2021-2-26). The C57 mice used in the production of mouse fibroblasts were obtained from the animal breeding room of our laboratory. The teratoma nod-SCID mice used in the differentiation test were purchased from Beijing Wetong Lihua Co., LTD., Beijing, China, and tested in our laboratory’s temporary animal breeding room.

### 4.2. Bovine Embryo Culture

The bovine in vitro fertilization and embryo culture are described in our previous report [11]. In short, bovine ovaries were obtained from a local abattoir, and oocytes were cultured for 22–24 h after maturation and fertilized with frozen cattle sperm. Six hours after fertilization, the eggs were transferred to the embryonic culture medium. When the embryos had developed to the 8-cell stage at 48 h, MM-102, PD0325901 and CHIR99021 (2i), or MM-102 plus PD0325901 and CHIR99021 (3i), was added to the medium. The embryos were then cultured for 7–8 days to develop into blastocysts.

### 4.3. Derivation of bESC Cell Lines

The bovine blastocysts obtained via in vitro fertilization were removed from the zona pellucida and plated in 4-well plates, and mTeSR-E6 (05946, STEMCELL Vancouver, Technologies, Canada) was added with 20 ng/mL FGF-2 (100-18B, STEMCELL Technologies, Vancouver, BC, Canada), 2.5 μM IWR-1 (I0161, Sigma-Aldrich, Shanghai, China) and 0.1328 g/mL low fatty acid BSA (219989925, MP Biomedicals, Auckland, NZ, USA), and were incubated at 37 °C and 5% CO_2_. After 48–72 h, ICMs adhered to the feeder layer, and the medium was changed daily. Outgrowths (after 8–10 days in culture) were dissociated and passaged using TrypLE (12563011, Gibco, CA, USA) and were reseeded in the presence of 10 μM Rho kinase (ROCK) inhibitor Y-27632 (SCM075, Sigma, Shanghai, China). The established bESC lines were grown in 12-well plates and were passaged every 5 days at a 1:4–1:6 split ratio. To increase cell survival, the ROCK inhibitor Y-27632 (10 μM) was added to the wells 1 h before passaging and was also added to the newly prepared wells containing MEFs treated with mitomycin and fresh culture medium during the first 24 h of cultivation.

### 4.4. Alkaline Phosphatase (AP) Staining

Prior to AP staining, the bESCs were cultured on feeder cells for 4 days. Following the removal of the culture medium, the cells were washed with DPBS and fixed for 10 min at room temperature using 4% paraformaldehyde. The fixed cells were washed three times with DPBS and stained for 1–3 h at room temperature in the dark, using the BCIP/NBT Color Development Substrate Kit (C3206, Beyotime Biotechnology, Shanghai, China). The cells were washed with DPBS to terminate the staining reaction and were subsequently maintained in DPBS.

### 4.5. Karyotype Analysis

During the flourishing period of cell division, which is usually on the 4th day of culture, 2 mg/mL colchicine (64-86-8, Sigma, Shanghai, China) was added to the culture medium and incubated at 37 °C for 2.5 h. Then, the cells were treated with 8 mL 0.075 mol KCl solution in a 37 °C water bath for 30 min. We collected the cells and added 1mL fixing solution (acetic acid: methanol = 1:3), and the solution was mixed. After centrifugation at 1500 rpm for 5 min, the supernatant was discarded, 10 mL fixing solution was added and mixed, and the cells stood at room temperature for 15min. After carrying out these fixing procedures three times, 50–200 μL fixing solution was added to the cells and mixed well, according to the amount of precipitation. Then, the cells were dropped from a height of approximately 1 m onto a pre-cooled slide at −20 °C, and left to dry overnight. The prepared sample was immersed in 0.025% trypsin solution preheated at 37 °C, and digested for 10 s; then, it was quickly washed twice in 0.85% NaCl solution, and dried. The sample was dyed in Jimsa solution (48,900, Sigma, Shanghai, China) for 15 min; then, we slowly washed off the excess with water, and dried it. Photographs were taken and analyzed using a cytogenetics workstation.

### 4.6. Quantitative Real-Time PCR (qRT-PCR)

The quantification of mRNAs was conducted via real-time PCR using specific primers. Real-time PCR was performed using an Applied Biosystems 7500 sequence detection system (Thermo Fischer Scientific, Shanghai, China) and KAPA SYBR^®^ FAST Universal qPCR master mix (Kapa Biosystems Pty, Boston, MA, USA). The PCR samples were analyzed in 96-well plates. Each reaction (20 mL) contained forward and reverse primers at 0.2 mM and 10 mL SYBR Green PCR master mix. The PCR steps included incubation for 5 min at 95 °C, followed by 40 cycles of 95°C for 10 s, 60 °C for 20 s, and 72 °C for 30 s. All reactions were performed at least in triplicate, and product identity was confirmed via melting curve analysis. Relative expression levels were determined using the 2^−ΔΔCT^ method, and normalized against GAPDH levels.

### 4.7. Statistical Analysis

Statistical analyses of qPCR results were performed using GraphPad Prism 8.3.0, and statistically significant differences between groups were identified using a paired *t*-test, with bESCs-102, bESCs-3i, bESCs-102-5, and bESCs-102-50 as the treatment groups and bESCs-F7 as the control group. The blastocyst rates, TE, ICM and total cells of the blastocysts subjected to different treatments were analyzed via a chi-squared test with Yates’ correction. The analyses were performed using the statistical software GraphPad PRISM 6.0 (GraphPad Software, Inc., La Jolla, CA, USA), and the results are presented as the means ± SDs. *p*-values less than 0.05 were considered statistically significant.

### 4.8. RNA-seq Analysis

Transcriptome (RNA) sequencing was performed by Tianjin Nuohe Zhiyuan Biotechnology Co., LTD, Tianjin, China. The PCA plot and volcano plot in this paper were generated using ggplot2 (version: 3.3.5) in R software 4.1.0, and the heatmap was generated using pheatmap (version:1.0.12); all track data (BS-seq, Chip-seq) were transformed to bigwig format used deeptools (version:3.4.3) [22], and then visualized using the Integrative Genomics Viewer (IGV, 2.14.1) [23].

### 4.9. Immunofluorescence Staining (IF)

Cells were cultured on coverslips on a 4-well plate. When grown to a suitable density, the cells were fixed with 4% paraformaldehyde for 10 min, and then treated with Triton X-100 to penetrate the cell membrane. After three washes with PBS, the cells were incubated with the primary antibody at 4 °C overnight. Then, the cells were incubated with the secondary antibody at room temperature for 1 h. After the removal of antibodies, the cells were incubated at room temperature with DAPI for 5 min. The slide was sealed after a microscopic examination. The antibodies used in this experiment included anti-OCT4 (sc-5279, Santa Cruz Biotechnology, Shanghai, China), anti-SOX2 (L1D6A2, Cell Signaling, Boston, MA, USA), anti-NANOG (500-P236, Peprotech, Shanghai, China), anti-SSEA-1 (MAB4301, Sigma, Shanghai, China), anti-SSEA-4 (MAB4304, Millipore, Shanghai, China), anti-TRA-1-60 (MAB4360, Shanghai, China), anti-TRA-1-81 (MAB4381, Shanghai, China), anti-AFP (AF5369, R&D Systems, Minneapolis, MN, USA), anti-SMA (ab5694, Abcam, Cambridge, MU, USA) and anti-GFAP (I1044, DAKO, Copenhagen, Denmark). 

### 4.10. Flow Cytometry Analysis

The cells were fixed with 4% paraformaldehyde for 10 min and permeabilized with 0.8% Triton X-100 for 10 min. Then, the cells were incubated with antibodies in 10% goat serum for 1h at room temperature. The primary antibody was incubated at 4 °C overnight and the secondary antibody was incubated at room temperature for 1 h. The DPBS was used to wash off the antibodies between each step. After incubation, the cells were screened and analyzed via flow cytometry (Cytoflex LX), according to the corresponding fluorescence intensity. The antibodies used in this experiment included anti-SSEA-1 (MAB4301, Sigma) and anti-SSEA-4 (MAB4304, Millipore).

### 4.11. Western Blotting (WB)

Cells were collected and protein was extracted using a Mammalian Protein Extraction Reagent (CWBIO, Taizhou, Jiangsu, China), according to the manufacturer’s procedure. A cracking product was added to the loading buffer and boiled in boiling water for 5 min for denaturation. We added the same amount of protein sample into the hole of the SDS-PAGE gel, and maintained a constant pressure of 90–120 V for 30 min–3 h. After the marker was separated, the gel was cut according to the marker position to obtain the corresponding protein sizes, and transferred to a nitrocellulose membrane with a constant current of 200 mA for 30 min–1 h. After the transfer of the required strip was completed, we placed it in 5% sealing solution and placed it at room temperature for 1 h. We transferred the membrane to the diluent of the primary antibody at 4 °C overnight, and washed the membrane 3 times with TBST solution. The secondary antibody was incubated at room temperature for 1 h, and TBST was used to wash the membrane 3 times; it was exposed to an indicator after the treatment. The antibodies used in this experiment included anti-H3 (4620S, Cell Signaling Technology, Boston, MA, USA), anti-H3K4me1 (ab8895, Abcam, Cambridge, MU, USA), anti-H3K4me2 (ab32356, Abcam), anti-H3K27me3 (ab6002, Abcam), anti-GAPDH (10494-1-AP, Proteintech, Franklin Lakes, NJ, USA), anti-PRDM14 (ab187881, Abcam), anti-P300 (ab54984, Abcam), anti-G9A (ab18894, Abcam), anti-DNMT3A (D23G1, Cell Signaling Technology, Boston, MA, USA) and anti-DNMT3B (bs-0301R, Bioss, Beijing, China). Images were obtained using a Tanon 5200 Multi Automatic Fluorescence and Chemiluminescence Imaging System (Tanon, Shanghai, China).

### 4.12. Formation of Embryoid Bodies (EB) In Vitro

When the density of the cells reached 80%, the cells were digested and collected. The differentiation medium from the first stage (90% IMDM + 10% FBS) was used to suspend the cells in low-adherent 35 mm culture dishes. The embryoid body formed at 6–7 days. The embryoid bodies were collected and centrifuged. Cells were resuspended in the second-stage differentiation medium (90%DMEM + 10% FBS) and plated into four-well plates for adherent growth. After 3 weeks of culture, the differentiated cells were identified via immunofluorescence staining, with the specific antibodies of three germ layers. The antibodies used in this experiment included anti-α-SMA (ab244177, Abcam, Cambridge, MU, USA), anti-AFP (MAB1368, R&D Systems, Minneapolis, MN, USA) and anti-GFAP (Z0334, DAKO, Copenhagen, Denmark).

### 4.13. Teratoma Formation In Vivo

The cells were collected and suspended in an appropriate amount of DPBS, and approximately 1 × 10^7^ cells were injected subcutaneously into Nod scid mice at each site. When the tumors were visible, the teratomas were removed from the mice. The resulting teratomas were paraffin-embedded, sectioned, and HE stained for analysis.

### 4.14. ChIP-seq Analysis

The Simple ChIP^®^ Plus Enzymatic Chromatin IP Kit (Magnetic Beads) (9005, Cell Signaling) was utilized to extract DNA fragments from the cells. The sequencing was performed by Tianjin Nuohe Zhiyuan Biotechnology Co., LTD, Tianjin, China. Illumina reads were first mapped to the UCSC bosTau9 reference using bwa-mem (version:0.7.17) [24] with default parameters. Next, Picard (http://broadinstitute.github.io/picard/, accessed on 12 June 2023, version 2.23.1) was used to mark up the PCR duplicates. Then, we used macs2 (v2.2.7.1, -nomodel-broad-broad-cutoff 0.1-shift 0-gsize 2.7e9-keep-dup auto) [25] to call the peaks. H3K4me1 signals were normalized using the MA norm (version: 1.1.4) [26] method for the quantitative comparison of ChIP-seq data, and the significant differential peaks were determined as log10(*p*-value) < 0 and M-value > 1. Subsequently, the differential peaks were annotated to UCSC bosTau9 using the R package ChIP seeker (version: 1.28.3) [27] and TxDb.Btaurus.UCSC.bosTau9.refGene (version: 3.10.0).

### 4.15. Bisulfite Genomic Sequencing Analysis

All cell lines in this experiment were used to analyze the promoter methylation of *OCT4* and *NANOG* via bisulfite sequencing PCR. DNA treatment and methylation-specific PCR were executed using the ZYMO EZ DNA Methylation-Gold Kit (ZYMO RESEARCH) and Takara Ex Taq (Hot Start Version), according to the manufacturers’ protocols. The PCR products were ligated into a pEASY-T1 Cloning Vector (Trans Gen Biotech, Beijing, China) for methylation sequencing. At least 10 clones per gene were sequenced and analyzed for each sample.

### 4.16. Whole-Genome Methylation Sequencing

DNA methylation sequencing was performed by Tianjin Nuohe Zhiyuan Biotechnology Co., LTD. The R package edge R (version: 3.34.1) [28] was used to analyze the differences between the treatments and the control. The read counts were tested for differential expression using the ‘exact test’. The differentially expressed genes (DEGs) in the data set with |log2 (fold change)|≥ 1.5 and adjusted *p* ≤ 0.05 were selected for the subsequent analyses. Next, the cluster profiler (version: 4.0.5) [29] package was used to annotate and enrich the GO and KEGG pathways for DEGs. *p* ≤ 0.05 was determined as a cut-off criterion for significant enrichment.

## Figures and Tables

**Figure 1 ijms-24-11901-f001:**
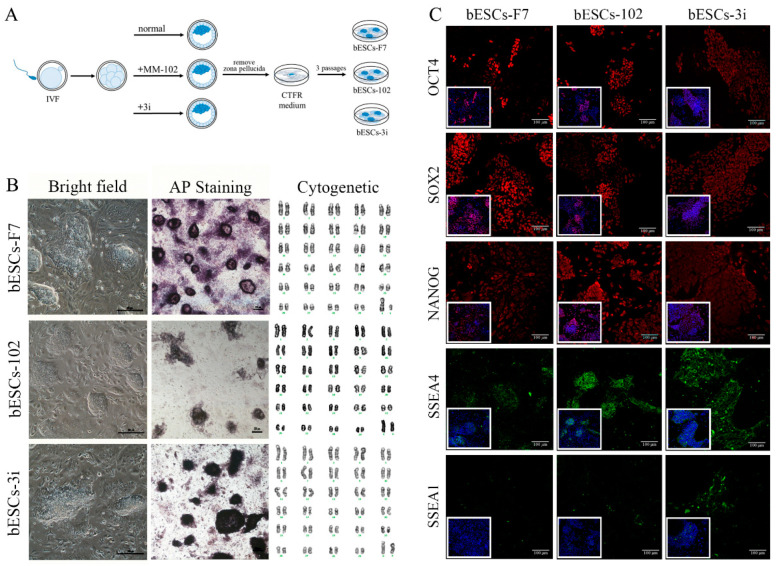
Generation of bovine blastocysts and establishment of bESCs in CTFR. (**A**) Schematic diagram of embryo culture with different combinations of MM-102, CHIR99021 and PD0325901, and establishment of bESCs from blastocysts by CTFR culture system. Note: 3i indicated CHIR99021 + PD3259012 + MM-102. (**B**) Morphology, alkaline phosphatase staining and karyotyping analysis of bESCs-F7, bESCs-102 and bESCs-3i (scale bar, 200 μm). (**C**) Immunofluorescence staining of pluripotency markers OCT4, SOX2 and NANOG, primed pluripotency marker SSEA4 and naïve pluripotency marker SSEA1 (scale bar, 100 μm).

**Figure 2 ijms-24-11901-f002:**
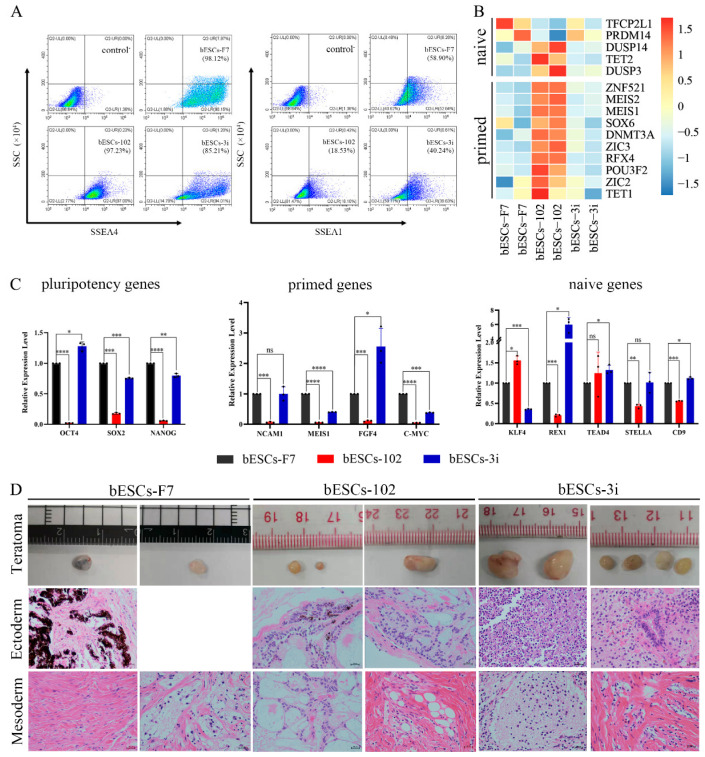
bESCs-102 and bESCs-3i have higher pluripotency marker expression than bESCs-F7. (**A**) Flow cytometry results showing expression of SSEA4 and SSEA1 in bESCs-F7, bESCs-102 and bESCs-3i. Negative control consisted of a mixture of three cell lines. (**B**) Transcriptome analysis of selected naïve and primed pluripotency markers in bESCs-F7, bESCs-102 and bESCs-3i. RNA-seq was performed, and RPKM values were used to define up-regulated genes (RPKM ≥ 1, red) and down-regulated genes (RPKM < 1, blue). (**C**) Relative expression level of pluripotency genes in bESCs-F7, bESCs-102 and bESCs-3i, determined via qRT-PCR. qRT-PCR data normalized to GAPDH. Data processing calculated using *t*-test; n = 3, ^ns^ *p* < 0.1, * *p* < 0.05, ** *p* < 0.01, *** *p* < 0.001, **** *p* < 0.0001. (**D**) Representative images showing H&E staining of histological sections derived from teratomas generated by bESCs-F7, bESCs-102 and bESCs-3i. Derived teratomas contained tissues of two germ lineages: ectoderm and mesoderm (scale bar, 100 μm).

**Figure 3 ijms-24-11901-f003:**
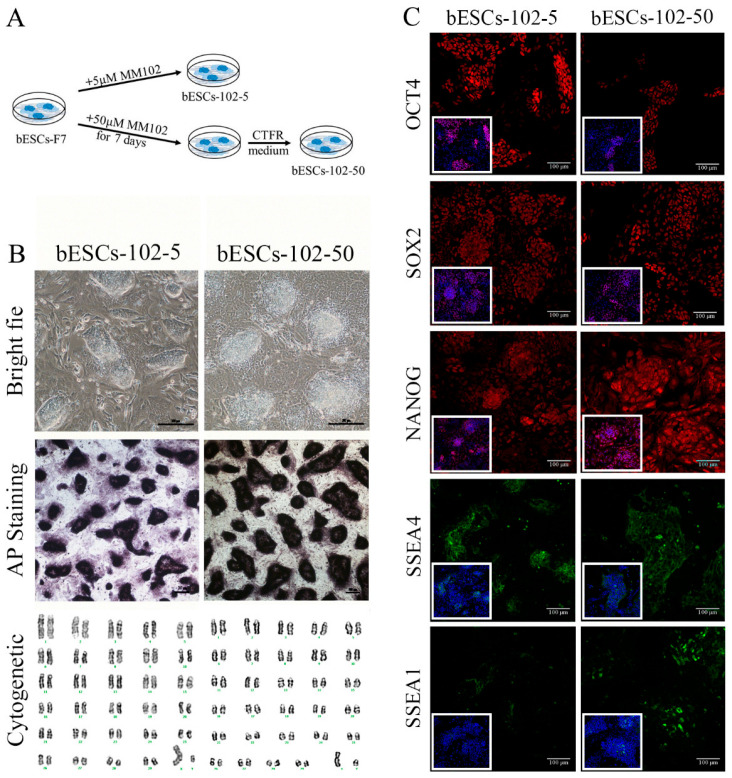
The establishment of MLL1 inhibited bESCs. (**A**) Schematic diagram of the establishment of MLL1-inhibited cell lines bESCs-102-5 and bESCs-102-50. (**B**) Morphology, alkaline phosphatase staining and karyotyping analysis of bESCs-102-5 and bESCs-102-50 (scale bar, 200 μm). (**C**) Immunofluorescence staining of pluripotent markers OCT4, SOX2 and NANOG, primed pluripotent marker SSEA4 and naïve pluripotent marker SSEA1 (scale bar, 100 μm).

**Figure 4 ijms-24-11901-f004:**
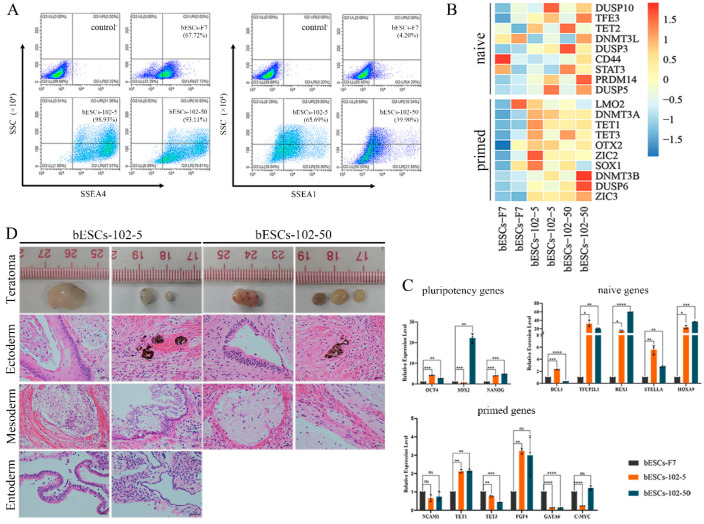
MLL1 inhibition improves pluripotency of bESCs. (**A**) Flow cytometry results showing expression of SSEA4 and SSEA1 in bESCs-F7, bESCs-102-5 and bESCs-102-50. Negative control consisted of a mixture of three cell lines. (**B**) Transcriptome analysis of selected naïve and primed pluripotent markers in different bESCs. RNA-seq was performed, and RPKM values were used to define up-regulated genes (RPKM ≥ 1; red) and down-regulated genes (RPKM < 1; blue). (**C**) Relative expression level of pluripotent genes in bESCs-F7, bESCs-102-5 and bESCs-102-50 by qRT-PCR. qRT-PCR data normalized to GAPDH. Data processing calculated using *t*-test; n = 3, ^ns^
*p* < 0.1, * *p* < 0.05, ** *p* < 0.01, *** *p* < 0.001, **** *p* < 0.0001. (**D**) H&E staining of histological sections derived from teratomas generated by bESCs-102-5 and bESCs-102-50. Derived teratomas contained tissues of all three germ layer lineages: ectoderm, mesoderm and endoderm (scale bar, 100 μm).

**Figure 5 ijms-24-11901-f005:**
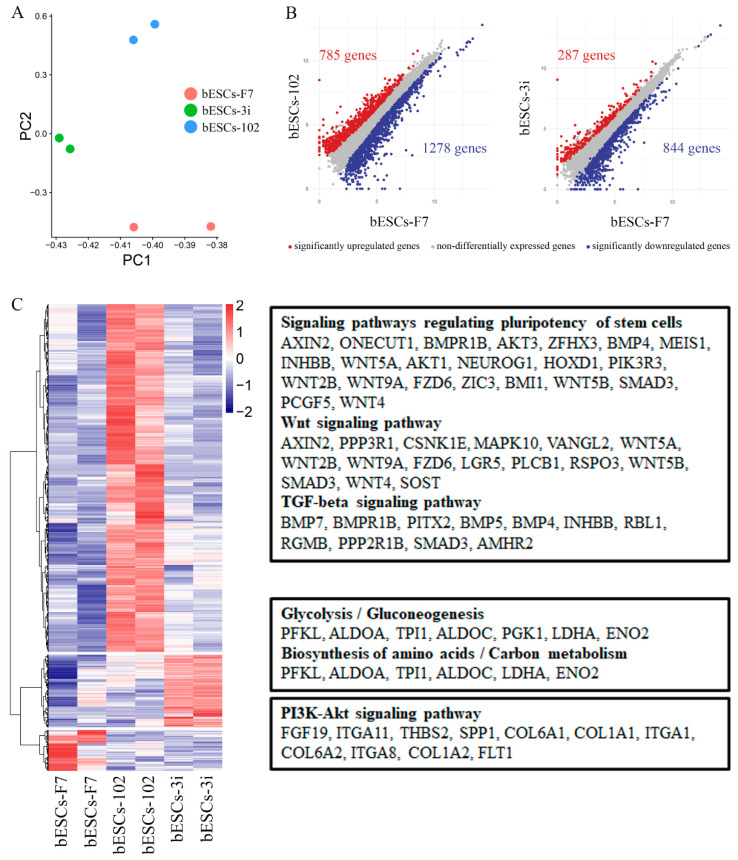
Comparison of pluripotency of bESCs-102 and bESCs-F7, and bESCs-3i and bESCs-F7. (**A**) PCA based on RNA-seq data of bESCs-F7, bESCs-102 and bESCs-3i. (**B**) Scatterplots showing significantly up-regulated (in red) and down-regulated (in blue) genes between bESC-F7 and bESC-102, and bESC-F7 and bESC-3i. Genes that were not differentially expressed are presented in gray. (**C**) Heat maps and KEGG analysis of highly expressed genes in bESCs-F7, bESCs-102 and bESCs-3i, respectively. RNA-seq was performed, and RPKM values were used to define up-regulated genes (RPKM ≥ 1, red) and down-regulated genes (RPKM < 1, blue).

**Figure 6 ijms-24-11901-f006:**
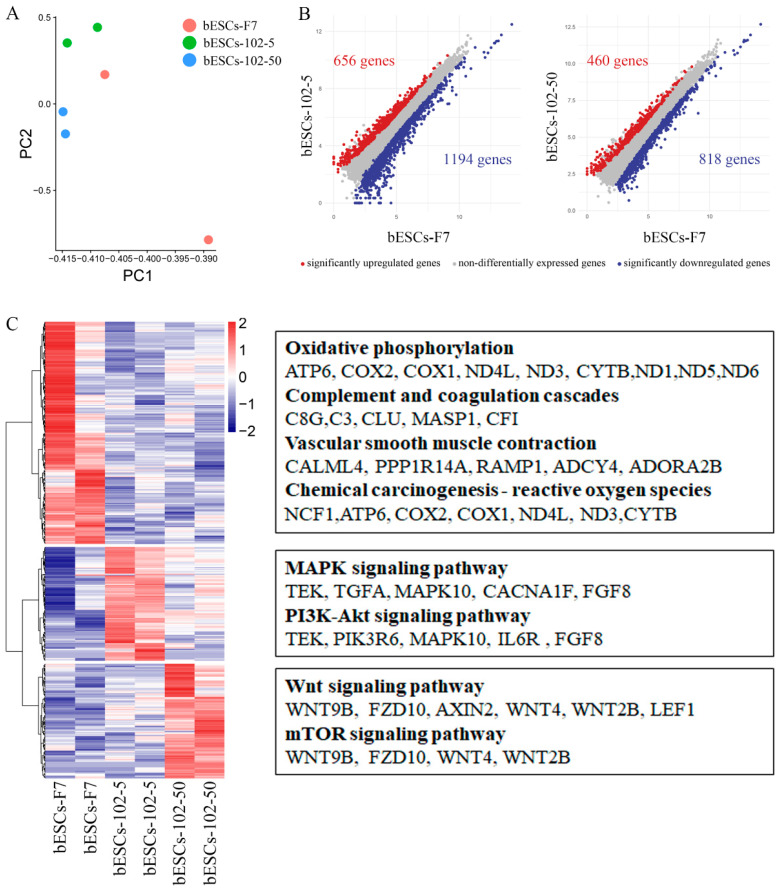
MLL1 inhibition enhanced pluripotent stem cell-related pathways in bESCs-102-5 and bESCs-102-50. (**A**) PCA based on RNA-seq data in bESCs-F7, bESCs-102-5 and bESCs-102-50. (**B**) Scatterplots show significantly up-regulated (in red) and down-regulated (in blue) genes between bESC-F7 and bESC-102-5, and bESC-F7 and bESC-102-50. Genes not differentially expressed are presented in gray. (**C**) Heat maps and KEGG analysis of highly expressed genes in bESCs-F7, bESCs-102-5 and bESCs-102-50, respectively. RNA-seq was performed, and RPKM values were used to define up-regulated genes (RPKM ≥ 0.5, red) and down-regulated genes (RPKM < 0.5, blue).

**Figure 7 ijms-24-11901-f007:**
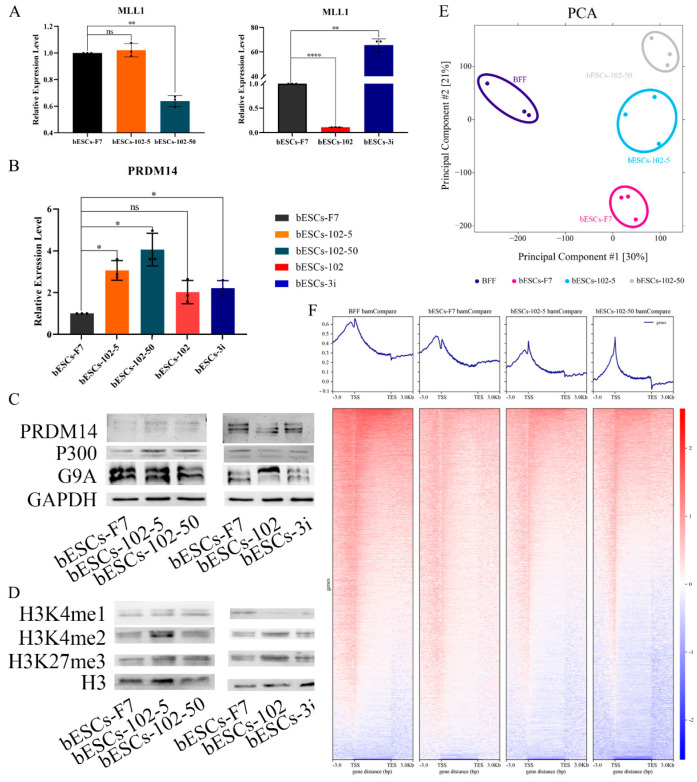
Histone modification of bESCs-F7 was changed following MLL1 inhibition. (**A**) Relative expression level of MLL1 in bESCs-F7, bESCs-102-5, bESCs-102-50 (**left**), and bESCs-102 and bESCs-3i (**right**) by qRT-PCR. qRT-qPCR data normalized to GAPDH. ^ns^ *p* < 0.1, * *p* < 0.05, ** *p* < 0.01, **** *p* < 0.0001. (**B**) Relative expression level of PRDM14 in bESCs-F7, bESCs-102-5, bESCs-102-50, bESCs-102 and bESCs-3i by qRT-qPCR. qRT-qPCR data normalized to GAPDH. Data processing calculated using *t*-test; n = 3, ^ns^ *p* < 0.1, * *p* < 0.05. (**C**) Western blotting showing protein levels of PRDM14, P300 and G9A in bESCs-F7, bESCs-102-5, bESCs-102-50 (**left**), and bESCs-102 and bESCs-3i (**right**). (**D**) Western blotting showing protein level of H3K3me1, H3K4me2 and H3K27me3 in bESCs-F7, bESCs-102-5, bESCs-102-50 (**left**), and bESCs-102 and bESCs-3i (**right**). (**E**) PCA based on ChIP-seq data of H3K4me1 in BFFs, bESCs-F7, bESCs-102-5 and bESCs-102-50. (**F**) Heatmap of ChIP-seq data of global genes of BFFs, bESCs-F7, bESCs-102-5 and bESCs-102-50. RPKM values were used to define up-regulated genes (RPKM ≥ 1, red) and down-regulated genes (RPKM < 1, blue).

**Figure 8 ijms-24-11901-f008:**
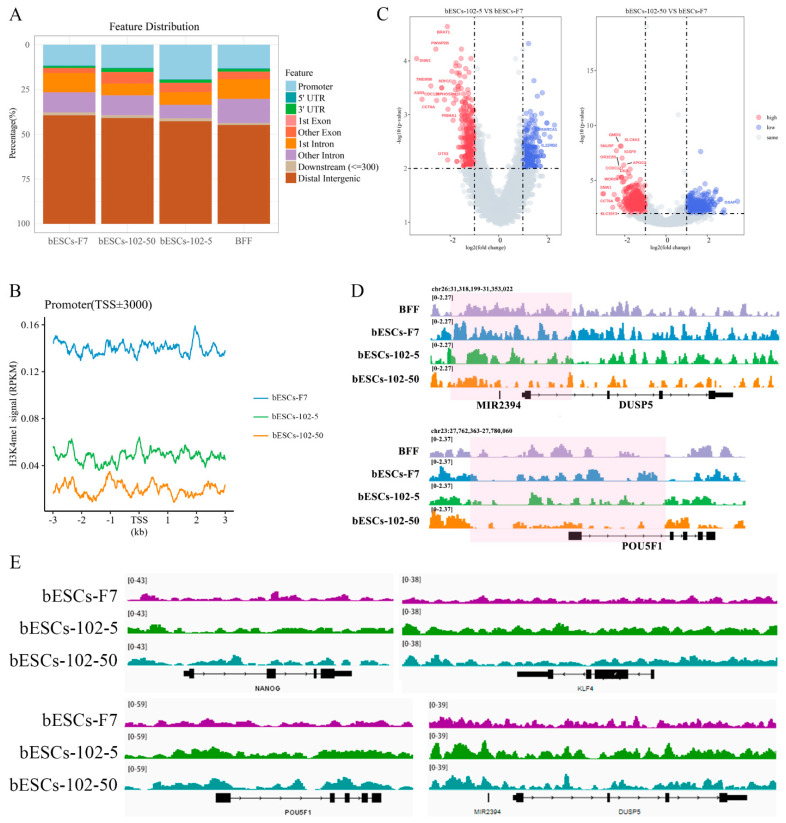
MLL1 inhibition changed the distribution of H3K4me1 at the promoter region. (**A**) Ratios of H3K4me1 in gene regulatory regions based on H3K4me1 ChIP-seq of bESCs-F7, bESCs-102-5, bESCs-102-50 and BFFs. (**B**) Comparison of H3K4me1 modification levels in the promoters of bESCs-F7, bESCs-102-5 and bESCs-102-50. (**C**) Volcano plot displaying H3K4me1 modification of the promoters of bESCs-F7, bESCs-102-5, and bESCs-102-50. Red and blue dots indicate up-regulated and down-regulated genes. (**D**) Peak map of H3K4me1 enrichment in the promoter region of DUSP5 and OCT4 of bESCs-F7, bESCs-102-5 and bESCs-102-50 (pink regions). (**E**) Integrative Genomics Viewer (IGV) Genome Browser views showing H3K4me1 tracks of pluripotent genes in bESCs-F7, bESCs-102-5 and bESCs-102-50.

**Figure 9 ijms-24-11901-f009:**
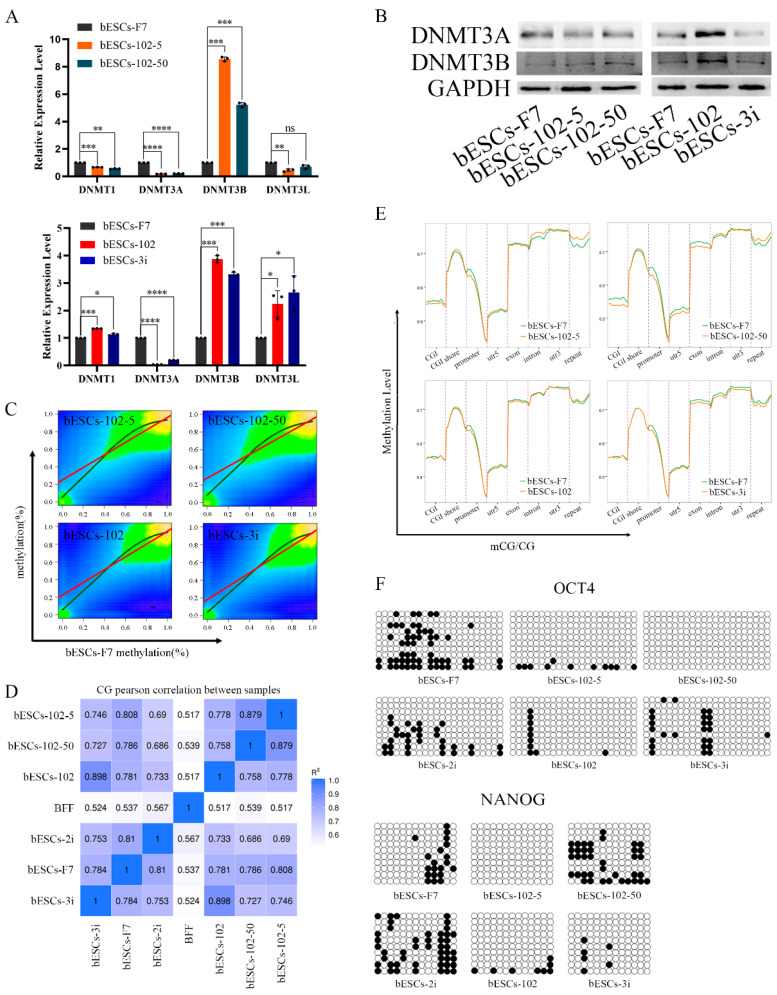
MLL1 inhibition altered DNA methylation in bESCs. (**A**) Relative expression level of DNMT family in bESCs-F7, bESCs-102-5, bESCs-102-50, bESCs-102 and bESCs-3i by qRT-PCR. qRT-qPCR data normalized to GAPDH. Data processing calculated using *t*-test; n = 3, ^ns^ *p* < 0.1, * *p* < 0.05, ** *p* < 0.01, *** *p* < 0.001, **** *p* < 0.0001. (**B**) Western blotting showing decreased expression of DNMT3A and DNMT3B in bESCs-F7, bESCs-102-5, bESCs-102-50, bESCs-102 and bESCs-3i. (**C**) Correlation plot of methylated sites in bESCs-F7 versus either bESCs-102-5, bESCs-102-50, bESCs-102 or bESCs-3i. Red line represents fit based on linear regression modeling (off-center best fit indicates lower correlation); blue line is based on LOESS weighted regression modeling (curved best-fit line indicates non-linear correlation). (**D**) Pearson correlation of bESCs-F7, bESCs-102-5, bESCs-102-50, bESCs-102 and bESCs-3i. (**E**) Comparison of DNA methylation of bESCs-F7, bESCs-102-5, bESCs-102-50, bESCs-102 and bESCs-3i shows differences in overall distribution of CG methylation levels on gene functional elements. (**F**) Bisulfite genomic sequencing of promoter regions of OCT4 and NANOG in bESCs-F7, bESCs-102-5, bESCs-102-50, bESCs-102 and bESCs-3i.

## Data Availability

Not available.

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
