# Peer review of "Mixed-Lineage Leukemia 1 Inhibition Enhances the Differentiation Potential of Bovine Embryonic Stem Cells by Increasing H3K4 Mono-Methylation at Active Promoters"

_ijms, 2023, doi:10.3390/ijms241511901_

Round 1

Reviewer 1 Report

In this original research report, Li et al describe the derivation and characterization of bovine embryonic stem cells (bESCs) following pre-treatment of IVF-produced bovine early embryos with a MLL1-inhibitor. The authors examine the changes in gene expression and in the epigenetic properties of the derived bESCs, such as histone H3K4 methylation patterns and genomic DNA methylation. As a result of the treatment with MLL1-inhibitor, the efficiency of bESC derivation was improved. In addition, the newly established bESCs show altered pluripotency gene expession and eigenetic properties, which correlated with improved in vitro and in vivo differentiation potential potential, as evidenced by the increased rate of teratoma formation. Similarly, treatment of earlier established bESCs leads to improved pluripotency. These changes can be explained by epigenetic changes in the promoter regions of pluripotency genes, such as reduced CG methylation in the promoters of pluripotency genes.

The manuscript contains a lot of interesting data and would be of significance to researchers working in the field of stem cells, particularly of farm species. The analysis of the ESCs is quite thorough and detailed. In general, I consider it acceptable for publication, as I have no major concerns, beside the small sample sizes (most experimental groups have n=2). However, I am aware that small sample sizes are common in this line of research, so the resilts of t-tests are accepted as long as they meet the criteria (Cohen’s D > 6 and normal distribution).

As a minor comment, I would recommend improving the materials and methods section where almost no information is given regarding the pre-treatment of the IVF embryos with MM-102 and the other small molecules (MM-102, 2i and 3i conditions). This information is given in the results section, but because the M&M section is usually the place where one finds it, it is a bit confusing not to see it there. Besides this, I have no further comments.

The text is easy to follow, but would benefit from minor editing of some of the expressions. Just to give one example, the sentence "Three cell lines have almost no endodermal derivatives in teratoma formation in vivo." (on page 7) would need to be edited, as it is not clear which three cell lines the authors mean. Another careful proofreading of the manuscript text to improve some of the sentence structure or fill in the missing punktuation (e.g. there is a missing period on page 8, end of paragraph 3) is therefore recommended.

Author Response

Dear reviewer:

Thank you very much for your very positive comments on our manuscript entitled "MLL1 inhibition enhances the differentiation potential of bovine embryonic stem cells by increasing H3K4 mono-methylation at active promoters". (ID: ijms-2476711). We have carefully gone through all your comments, and made corresponding modifications for our paper. We hope our responses are satisfied.

Our responses are listed here:

Comments and Suggestions for Authors

In this original research report, Li et al describe the derivation and characterization of bovine embryonic stem cells (bESCs) following pre-treatment of IVF-produced bovine early embryos with a MLL1-inhibitor. The authors examine the changes in gene expression and in the epigenetic properties of the derived bESCs, such as histone H3K4 methylation patterns and genomic DNA methylation. As a result of the treatment with MLL1-inhibitor, the efficiency of bESC derivation was improved. In addition, the newly established bESCs show altered pluripotency gene expession and eigenetic properties, which correlated with improved in vitro and in vivo differentiation potential potential, as evidenced by the increased rate of teratoma formation. Similarly, treatment of earlier established bESCs leads to improved pluripotency. These changes can be explained by epigenetic changes in the promoter regions of pluripotency genes, such as reduced CG methylation in the promoters of pluripotency genes.

The manuscript contains a lot of interesting data and would be of significance to researchers working in the field of stem cells, particularly of farm species. The analysis of the ESCs is quite thorough and detailed. In general, I consider it acceptable for publication, as I have no major concerns, beside the small sample sizes (most experimental groups have n=2). However, I am aware that small sample sizes are common in this line of research, so the resilts of t-tests are accepted as long as they meet the criteria (Cohen’s D > 6 and normal distribution).

As a minor comment, I would recommend improving the materials and methods section where almost no information is given regarding the pre-treatment of the IVF embryos with MM-102 and the other small molecules (MM-102, 2i and 3i conditions). This information is given in the results section, but because the M&M section is usually the place where one finds it, it is a bit confusing not to see it there. Besides this, I have no further comments.

Responses:

Thank you very much for your positive comments on our manuscript, we have put lots of efforts to this work and we will do our best to make it better according to your comments.

Comments #1

“In general, I consider it acceptable for publication, as I have no major concerns, beside the small sample sizes (most experimental groups have n=2). However, I am aware that small sample sizes are common in this line of research, so the resilts of t-tests are accepted as long as they meet the criteria (Cohen’s D > 6 and normal distribution).”

Responses:

We are sorry for the small sizes for some our experiments. Because the repeatability of most experiments is good, we did not do three repeats. We recalculated the results using t-test and annotated the corresponding P-value symbol on the corresponding histogram using t-test.

Comments #2

“As a minor comment, I would recommend improving the materials and methods section where almost no information is given regarding the pre-treatment of the IVF embryos with MM-102 and the other small molecules (MM-102, 2i and 3i conditions). This information is given in the results section, but because the M&M section is usually the place where one finds it, it is a bit confusing not to see it there. Besides this, I have no further comments.”

Responses:

We are sorry for this mistake. We have provided the pre-treatment of the IVF embryos with MM-102 and the other small molecules (MM-102, 2i and 3i conditions) in the materials and methods section in the revised manuscript. Detailed description is as follows:

Bovine embryo culture

Bovine in vitro fertilization and embryo culture was described in our previous report  (Han et al., 2020). In short, bovine ovaries were obtained from a local abattoir, and oocytes were cultured for 22-24h after the maturation and fertilized with the frozen cattle sperm. 6h after fertilization, the eggs were transferred to the embryonic culture medium. When embryos developed to 8-cell stage at 48h, MM-102, PD0325901 and CHIR99021 (2i), or MM-102 plus PD0325901 and CHIR99021 (3i) was added to the medium. The embryos were then cultured for 7-8 days to develop into blastocysts.  For details, please see P18.

Comments #3

The text is easy to follow, but would benefit from minor editing of some of the expressions. Just to give one example, the sentence "Three cell lines have almost no endodermal derivatives in teratoma formation in vivo." (on page 7) would need to be edited, as it is not clear which three cell lines the authors mean.

Responses:

We have gone through our paper again, and made corresponding changes for some of unclear expressions. The sentence "Three cell lines have almost no endodermal derivatives in teratoma formation in vivo." (on page 7) has been corrected to “bESCs-102, bESCs-2i and bESCs-3i have almost no endodermal derivatives in terato-ma formation in vivo.”

Comments #4

Another careful proofreading of the manuscript text to improve some of the sentence structure or fill in the missing punctuations (e.g. there is a missing period on page 8, end of paragraph 3) is therefore recommended.

Responses:

We have carefully gone through our paper again, and improved the sentence structure and filled in the missing punctuations. The missing periods on page 8, end of paragraph 3 has been added back.

We hope that our responses and modifications are satisfactory and that the manuscript is acceptable for publication. Thanks again for the hard work of the editor and reviewers!

Yours sincerely,

Xueling Li

Xueling Li, PhD

State Key Laboratory of Reproductive Regulation & Breeding of Grassland Livestock

Research Center for Laboratory Animal Science

Inner Mongolia University

No. 24 Zhao Jun Rd

Yu Quan Qu, Hohhot, Inner Mongolia, P. R. China 010070

Phone: +86 471 3679807

Fax: +86 471 3679807

E-mail: lixueling@imu.edu.cn

Reviewer 2 Report

Immunoblots (representative of how many experiments?) are not convincing. They must be cropped in a way that retains information about antigen size and antibody specificity. The cropped images must retain sufficient area around the band(s) of interest, including the positions of at least one actual molecular weight marker above and one below the band(s). Please prepare each run including at least two lanes for each experimental group. Please avoid overexposed bands (especially in the loading controls). Moreover, to ensure reproducibility by other laboratories, the manufacturer name and catalog number(s) of all the antibodies used should be clearly specified in the material and method section. Please provide uncropped gels as supplementary material showing visible MW markers.

Statistical analysis: I did not find details on the normal distribution of values (e.g. Kolmogorov-Smirnov Test) in order to apply the t test. Moreover, the test applied and the n should be clearly stated in each figure legend.

Bar graphs with error bars do not allow a direct evaluation of the distribution of the data. The authors should present their continuous data in scatter/dot plots (especially in case of a limited number of observations), showing the individual data points together with the average/error bars.

The discussion in its present form fails to interpret the data in the context of what is known in the field: it sounds somehow redundant, as it largely summarizes again data already presented in the Results without placing them in the proper scientific context.

The following pertinent reports should be mentioned/discussed:

PMID: 31951812

PMID: 36217550

PMID: 34693227

PMID: 31775036

PMID: 24887289

PMID: 37176093

PMID: 26996599

PMID: 36711238

PMID: 28702240

PMID: 21925392

PMID: 35128356

PMID: 34401663

The paper is mainly descriptive and focused on its (not fully supported) conclusions, not adequately acknowledging the limitations of the study. The strengths and limitations of the study should be deeply addressed, taking into account sources of potential bias or imprecision: Discuss both direction and magnitude of any potential bias.

English language (syntax, grammar, correct choice of words, correct use of adjectives and adverbs) needs significant editing throughout the text. Professional assistance must be sought.

The color palette is not red-green color blind friendly. Please redo the figures without red and green on the same figure.

A proper dimensional bar should be provided for each picture. 

It is advisable to the Authors to incorporate a pictorial or cartoon representation of the whole mechanism of the study to increase the overall impact of the manuscript.

English language (syntax, grammar, correct choice of words, correct use of adjectives and adverbs) needs significant editing throughout the text. 

Author Response

Dear reviewer:

Thank you for your comments on our manuscript entitled "MLL1 inhibition enhances the differentiation potential of bovine embryonic stem cells by increasing H3K4 mono-methylation at active promoters". (ID: ijms-2476711). We have carefully gone through all your comments, and made corresponding modifications for our paper. We hope our responses are satisfied.

Our responses are listed here:

Comments #1

(1) Immunoblots (representative of how many experiments?) are not convincing. (2) They must be cropped in a way that retains information about antigen size and antibody specificity. The cropped images must retain sufficient area around the band(s) of interest, including the positions of at least one actual molecular weight marker above and one below the band(s). (3) Please prepare each run including at least two lanes for each experimental group. Please avoid overexposed bands (especially in the loading controls). (4) Moreover, to ensure reproducibility by other laboratories, the manufacturer name and catalog number(s) of all the antibodies used should be clearly specified in the material and method section. (3) Please provide uncropped gels as supplementary material showing visible MW markers.

Responses:

Thank you very much for your very detailed comments about our Western blotting experiments here. (1) We have performed at least three times of Western blotting for each antibody, so the immunoblots representative of more than 3 times experiments. (2) We have recropped the bands in a way that retained information about antigen size and antibody specificity (please see Figure 7 C and D, Figure 9 B). The cropped images retained sufficient area around the band(s) of interest, including the positions of at least one actual molecular weight marker above and one below the band(s). (3) We have ensued each run including at least two lanes for each experimental group and avoid overexposed bands. The uncropped gels showing visible MW markers are provided in supplementary materials. (4) The manufacturer name and catalog number(s) of all the antibodies used are clearly specified in the Materials and Methods section (please see P20 Western Blotting (WB)).

Comments #2

Statistical analysis: I did not find details on the normal distribution of values (e.g. Kolmogorov-Smirnov Test) in order to apply the t test. Moreover, the test applied and the n should be clearly stated in each figure legend.

Responses:

We have recalculated the data using t-test by GraphPad Prism 8.3.0, and relevant changes were added and n was clearly stated in each figure legend. (Please see Figure 2, Figure 4, Figure 7 and Figure 9, Data processing calculated using paired t-test, n=3.) And detailed procedure was added to Materials and Methods:

Statistical Analysis

Statistical analyses of qPCR results were performed using the GraphPad Prism 8.3.0, and statistically significant differences between groups were identified using paired t-test, bESCs-102, bESCs-3i, bESCs-102-5, bESCs-102-50 as treatment groups, bESCs-F7 as control group. P-value less than 0.05 was considered statistically significant.

Comments #3

Bar graphs with error bars do not allow a direct evaluation of the distribution of the data. The authors should present their continuous data in scatter/dot plots (especially in case of a limited number of observations), showing the individual data points together with the average/error bars.

Responses:

The scatter plot has been superimposed on the column chart as suggested. For details, please see Figure 2, Figure 4, Figure 7 and Figure 9.

Comments #4

 The discussion in its present form fails to interpret the data in the context of what is known in the field: it sounds somehow redundant, as it largely summarizes again data already presented in the Results without placing them in the proper scientific context. The following pertinent reports should be mentioned/discussed:

PMID: 31951812

PMID: 36217550

PMID: 34693227

PMID: 31775036

PMID: 24887289

PMID: 37176093

PMID: 26996599

PMID: 36711238

PMID: 28702240

PMID: 21925392

PMID: 35128356

PMID: 34401663

Responses:

Thank you very much for your constructive suggestions here, which help us consider more about this study, and is very useful for rewetting our Discussion. We have downloaded and read all suggested papers that you kindly provided, and made massive changes to our Discussion. For details, please see discussion on Page17 3. DISSCUSSION. The details are as followings:

Given the highly accessible and hyperactive chromatin structures in pluripotent stem cells (PSCs), it is generally assumed that H3K4me plays an important ‘‘housekeeping’’ role in PSCs and is necessary for PSCs to maintain self-renewal and unlimited differentiation potential (De Los Angeles et al., 2015). Zhang et al. found that H3K4me1 was significantly different between mouse ESCs and EpiSCs, but not H3K4me3. Interestingly, inhibition of MLL1 led to genome-wide change of H3K4me1 in EpiSCs and global redistribution of H3K4me1 at enhancers and represses lineage determinant factors and EpiSC markers, which indirectly regulate the transcription of mouse ESCs (Zhang et al., 2016). Naive PSCs are distinguished from primed PSCs by the capacity to form a teratoma and a chimeric animal following introduction into pre-implantation embryos, and exhibited unique transcription makers and DNA methylation levels (De Los Angeles et al., 2015). In recent years, significant advances have been made to convert the primed state of hPSCs into naïve pluripotency. Human naïve pluripotency has been shown to differentiate better than the primed state, which is associated with the accumulation of DNA methylation and epigenetic repressive marks in primed state (Varzideh et al. 2023). Previous studies show that MLL1 inhibition transformed mouse EpiSCs into naïve pluripotent state (Zhang et al., 2016), and combined treatment of MLL1 inhibitor and 1α,25-dihydroxyvitamin D3 (1,25-(OH)2D3) cooperatively enhanced functionality of expanded PSCs, triggering an extended 2C-like state in vitro and robust totipotent-like property in vivo (Zhang et al. 2019), which indicated MLL1 inhibition increase the differentiation capacity of PSCs. In this study, we found that MLL1 inhibition improves the pluripotency and differentiation potentials of bESCs by decreasing H3K4me1 modification at promoters and altered the distribution of DNA methylation in bESCs. However, MLL1 inhibition increased some naïve and primed markers simultaneously, but not transform bESCs from primed to naïve state like in mouse EpiSCs. And we also found that MLL1 inhibition by itself cannot transform human ESCs to naïve state as well (data not shown). The reasons may lie in the following aspects: firstly, the MLL1 inhibitor used in our study is different from Dou’s lab (Zhang et al. 2016), which may lead some different effects; Secondly, MM-102 only up-regulated some of the stem cell signaling pathways such as PI3K-Akt and WNT, which may not enough to transform bESCs from primed to naive state; finally, the characteristics of PSCs from domestic animals are quite different from that of mouse and human, and further investigation is needed to thoroughly define PSCs of domestic animals.

MLL1 epigenetically regulates gene expression patterns that specify cellular identity in both embryonic development and adult stem cell populations. Mll1 is required for the expression of neurogenic – but not gliogenic – transcriptional modules in a multipotent neural stem cell (NSC) population and further indicate that specific Mll1-dependent genes may be useful for direct reprogramming strategies (Potts et al. 2014). MLL1-induced activation of a Rac/Rho/Integrin signaling axis to enhance hematopoiesis from murine embryonic stem cells (Yang et al. 2020). In this study, MLL1 inhibition in bESCs increased the expression PRDM14, which is a histone lysine methyltransferase and also a common marker of primordial germ cells and pluripotent embryonic stem cells (Nady et al., 2015). Moreover, inhibition of MLL1 down-regulated the expression level of DNMT3A, up-regulated the expression level of DNMT3B. DNMT3B is the primary driver of de novo DNA methylation on actively transcribed genes, while DNMT3A plays a minimal role in ESCs (Baubec et al., 2015). These results indicated the effects of MLL1 inhibition in bESCs mainly on epigenetic level. MLL1 inhibition changed the distribution of DNA methylation, which was mainly reflected in the increase of the overall CpG methylation level, the decrease of mCG in the upstream and downstream regions of the genes, and the decrease of mCG in the promoter region. Increased expression of DNMT3B in bESCs is associated with increased DNA methylation (CpG). While mCG in the promoter region of pluripotent genes such as OCT4 and NANOG decreased upon MLL1 inhibition by bisulfite genomic sequencing of the promoter regions. These changes may be one of the reasons why MM-102 changes the pluripotency of bESCs. Although the distribution of DNA methylation was changed by MLL1 inhibition, the global DNA hypomethylation observed in naïve PSCs did not obtain. To achieve the real naïve PSCs for bovine, the thorough understanding the control of puripotency of ruminants is needed.

Comments #5

The paper is mainly descriptive and focused on its (not fully supported) conclusions, not adequately acknowledging the limitations of the study. The strengths and limitations of the study should be deeply addressed, taking into account sources of potential bias or imprecision: Discuss both direction and magnitude of any potential bias.

Responses:

Thank you for your very helpful suggestions here. We have deeply thought about the strengths and limitations of our study, and considered the sources of potential bias or imprecision and discussed both direction and magnitude of any potential bias in the Discussion section. For details, please see discussion on Page17 3. DISSCUSSION.  

Comments #6

The color palette is not red-green color blind friendly. Please redo the figures without red and green on the same figure.

Responses:

The red and green in the figure 2 C, figure 7 A and B and figure 9 A have been replaced by red and blue.

Comments #7

A proper dimensional bar should be provided for each picture.

Responses:

Appropriate size bars have been provided for each image. (Please see Figure 2 D, Figure 4 D.)

Comments #8

 It is advisable to the Authors to incorporate a pictorial or cartoon representation of the whole mechanism of the study to increase the overall impact of the manuscript.

Responses:

Cartoon representation of the whole mechanism of the study is submitted separately as a Graphical Abstract.

We hope that our responses and modifications are satisfactory and that the manuscript is acceptable for publication. Thanks again for the hard work of the editor and reviewers!

We hope that our responses and modifications are satisfactory and that the manuscript is acceptable for publication. Thanks again for the hard work of the editor and reviewers!

Yours sincerely,

Xueling Li

Xueling Li, PhD

State Key Laboratory of Reproductive Regulation & Breeding of Grassland Livestock

Research Center for Laboratory Animal Science

Inner Mongolia University

No. 24 Zhao Jun Rd

Yu Quan Qu, Hohhot, Inner Mongolia, P. R. China 010070

Phone: +86 471 3679807

Fax: +86 471 3679807

E-mail: lixueling@imu.edu.cn

Supplement:

Original Immunoblots for Western blotting

PRDM14, P300, G9A, GAPDH for figure 7C

H3K4me1, H3K4me2, H3K27me3 and H3 for figure 7D

DNMT3A, DNMT3B,GAPDH for figure 9B

Cells were collected and protein was extracted by Mammalian Protein Extraction Reagent (CWBIO, China) according to the manufacturer’s procedure. The product is added to the Loading Buffer and boiled in boiling water for 5 minutes for denaturation. Load the same amount of protein sample into SDS-PAGE gel and electrophoresis at 90 - 120 V for 30 min - 3 h. After maker separated, the gel is cut according to the marker position to obtain corresponding protein sizes, and transferred to nitrocellulose membrane, at constant current 200 mA for 30 min-1 h. After the transfer of the required strip is completed, place it in 5% sealing solution and place it at room temperature for 1 h. Transfer the membrane to the diluent of primary antibody at 4℃ and incubated overnight, and then wash the membrane with TBST solution for 3 times. The membrane was incubated with second antibody at room temperature for 1 h, and washed with TBST for 3 times. After treatment the membrane is exposed in the indicator. The anti-bodies used in

Dear reviewer:

Thank you for your comments on our manuscript entitled "MLL1 inhibition enhances the differentiation potential of bovine embryonic stem cells by increasing H3K4 mono-methylation at active promoters". (ID: ijms-2476711). We have carefully gone through all your comments, and made corresponding modifications for our paper. We hope our responses are satisfied.

Our responses are listed here:

Comments #1

(1) Immunoblots (representative of how many experiments?) are not convincing. (2) They must be cropped in a way that retains information about antigen size and antibody specificity. The cropped images must retain sufficient area around the band(s) of interest, including the positions of at least one actual molecular weight marker above and one below the band(s). (3) Please prepare each run including at least two lanes for each experimental group. Please avoid overexposed bands (especially in the loading controls). (4) Moreover, to ensure reproducibility by other laboratories, the manufacturer name and catalog number(s) of all the antibodies used should be clearly specified in the material and method section. (3) Please provide uncropped gels as supplementary material showing visible MW markers.

Responses:

Thank you very much for your very detailed comments about our Western blotting experiments here. (1) We have performed at least three times of Western blotting for each antibody, so the immunoblots representative of more than 3 times experiments. (2) We have recropped the bands in a way that retained information about antigen size and antibody specificity (please see Figure 7 C and D, Figure 9 B). The cropped images retained sufficient area around the band(s) of interest, including the positions of at least one actual molecular weight marker above and one below the band(s). (3) We have ensued each run including at least two lanes for each experimental group and avoid overexposed bands. The uncropped gels showing visible MW markers are provided in supplementary materials. (4) The manufacturer name and catalog number(s) of all the antibodies used are clearly specified in the Materials and Methods section (please see P20 Western Blotting (WB)).

Comments #2

Statistical analysis: I did not find details on the normal distribution of values (e.g. Kolmogorov-Smirnov Test) in order to apply the t test. Moreover, the test applied and the n should be clearly stated in each figure legend.

Responses:

We have recalculated the data using t-test by GraphPad Prism 8.3.0, and relevant changes were added and n was clearly stated in each figure legend. (Please see Figure 2, Figure 4, Figure 7 and Figure 9, Data processing calculated using paired t-test, n=3.) And detailed procedure was added to Materials and Methods:

Statistical Analysis

Statistical analyses of qPCR results were performed using the GraphPad Prism 8.3.0, and statistically significant differences between groups were identified using paired t-test, bESCs-102, bESCs-3i, bESCs-102-5, bESCs-102-50 as treatment groups, bESCs-F7 as control group. P-value less than 0.05 was considered statistically significant.

Comments #3

Bar graphs with error bars do not allow a direct evaluation of the distribution of the data. The authors should present their continuous data in scatter/dot plots (especially in case of a limited number of observations), showing the individual data points together with the average/error bars.

Responses:

The scatter plot has been superimposed on the column chart as suggested. For details, please see Figure 2, Figure 4, Figure 7 and Figure 9.

Comments #4

 The discussion in its present form fails to interpret the data in the context of what is known in the field: it sounds somehow redundant, as it largely summarizes again data already presented in the Results without placing them in the proper scientific context. The following pertinent reports should be mentioned/discussed:

PMID: 31951812

PMID: 36217550

PMID: 34693227

PMID: 31775036

PMID: 24887289

PMID: 37176093

PMID: 26996599

PMID: 36711238

PMID: 28702240

PMID: 21925392

PMID: 35128356

PMID: 34401663

Responses:

Thank you very much for your constructive suggestions here, which help us consider more about this study, and is very useful for rewetting our Discussion. We have downloaded and read all suggested papers that you kindly provided, and made massive changes to our Discussion. For details, please see discussion on Page17 3. DISSCUSSION. The details are as followings:

Given the highly accessible and hyperactive chromatin structures in pluripotent stem cells (PSCs), it is generally assumed that H3K4me plays an important ‘‘housekeeping’’ role in PSCs and is necessary for PSCs to maintain self-renewal and unlimited differentiation potential (De Los Angeles et al., 2015). Zhang et al. found that H3K4me1 was significantly different between mouse ESCs and EpiSCs, but not H3K4me3. Interestingly, inhibition of MLL1 led to genome-wide change of H3K4me1 in EpiSCs and global redistribution of H3K4me1 at enhancers and represses lineage determinant factors and EpiSC markers, which indirectly regulate the transcription of mouse ESCs (Zhang et al., 2016). Naive PSCs are distinguished from primed PSCs by the capacity to form a teratoma and a chimeric animal following introduction into pre-implantation embryos, and exhibited unique transcription makers and DNA methylation levels (De Los Angeles et al., 2015). In recent years, significant advances have been made to convert the primed state of hPSCs into naïve pluripotency. Human naïve pluripotency has been shown to differentiate better than the primed state, which is associated with the accumulation of DNA methylation and epigenetic repressive marks in primed state (Varzideh et al. 2023). Previous studies show that MLL1 inhibition transformed mouse EpiSCs into naïve pluripotent state (Zhang et al., 2016), and combined treatment of MLL1 inhibitor and 1α,25-dihydroxyvitamin D3 (1,25-(OH)2D3) cooperatively enhanced functionality of expanded PSCs, triggering an extended 2C-like state in vitro and robust totipotent-like property in vivo (Zhang et al. 2019), which indicated MLL1 inhibition increase the differentiation capacity of PSCs. In this study, we found that MLL1 inhibition improves the pluripotency and differentiation potentials of bESCs by decreasing H3K4me1 modification at promoters and altered the distribution of DNA methylation in bESCs. However, MLL1 inhibition increased some naïve and primed markers simultaneously, but not transform bESCs from primed to naïve state like in mouse EpiSCs. And we also found that MLL1 inhibition by itself cannot transform human ESCs to naïve state as well (data not shown). The reasons may lie in the following aspects: firstly, the MLL1 inhibitor used in our study is different from Dou’s lab (Zhang et al. 2016), which may lead some different effects; Secondly, MM-102 only up-regulated some of the stem cell signaling pathways such as PI3K-Akt and WNT, which may not enough to transform bESCs from primed to naive state; finally, the characteristics of PSCs from domestic animals are quite different from that of mouse and human, and further investigation is needed to thoroughly define PSCs of domestic animals.

MLL1 epigenetically regulates gene expression patterns that specify cellular identity in both embryonic development and adult stem cell populations. Mll1 is required for the expression of neurogenic – but not gliogenic – transcriptional modules in a multipotent neural stem cell (NSC) population and further indicate that specific Mll1-dependent genes may be useful for direct reprogramming strategies (Potts et al. 2014). MLL1-induced activation of a Rac/Rho/Integrin signaling axis to enhance hematopoiesis from murine embryonic stem cells (Yang et al. 2020). In this study, MLL1 inhibition in bESCs increased the expression PRDM14, which is a histone lysine methyltransferase and also a common marker of primordial germ cells and pluripotent embryonic stem cells (Nady et al., 2015). Moreover, inhibition of MLL1 down-regulated the expression level of DNMT3A, up-regulated the expression level of DNMT3B. DNMT3B is the primary driver of de novo DNA methylation on actively transcribed genes, while DNMT3A plays a minimal role in ESCs (Baubec et al., 2015). These results indicated the effects of MLL1 inhibition in bESCs mainly on epigenetic level. MLL1 inhibition changed the distribution of DNA methylation, which was mainly reflected in the increase of the overall CpG methylation level, the decrease of mCG in the upstream and downstream regions of the genes, and the decrease of mCG in the promoter region. Increased expression of DNMT3B in bESCs is associated with increased DNA methylation (CpG). While mCG in the promoter region of pluripotent genes such as OCT4 and NANOG decreased upon MLL1 inhibition by bisulfite genomic sequencing of the promoter regions. These changes may be one of the reasons why MM-102 changes the pluripotency of bESCs. Although the distribution of DNA methylation was changed by MLL1 inhibition, the global DNA hypomethylation observed in naïve PSCs did not obtain. To achieve the real naïve PSCs for bovine, the thorough understanding the control of puripotency of ruminants is needed.

Comments #5

The paper is mainly descriptive and focused on its (not fully supported) conclusions, not adequately acknowledging the limitations of the study. The strengths and limitations of the study should be deeply addressed, taking into account sources of potential bias or imprecision: Discuss both direction and magnitude of any potential bias.

Responses:

Thank you for your very helpful suggestions here. We have deeply thought about the strengths and limitations of our study, and considered the sources of potential bias or imprecision and discussed both direction and magnitude of any potential bias in the Discussion section. For details, please see discussion on Page17 3. DISSCUSSION.  

Comments #6

The color palette is not red-green color blind friendly. Please redo the figures without red and green on the same figure.

Responses:

The red and green in the figure 2 C, figure 7 A and B and figure 9 A have been replaced by red and blue.

Comments #7

A proper dimensional bar should be provided for each picture.

Responses:

Appropriate size bars have been provided for each image. (Please see Figure 2 D, Figure 4 D.)

Comments #8

 It is advisable to the Authors to incorporate a pictorial or cartoon representation of the whole mechanism of the study to increase the overall impact of the manuscript.

Responses:

Cartoon representation of the whole mechanism of the study is submitted separately as a Graphical Abstract.

We hope that our responses and modifications are satisfactory and that the manuscript is acceptable for publication. Thanks again for the hard work of the editor and reviewers!

We hope that our responses and modifications are satisfactory and that the manuscript is acceptable for publication. Thanks again for the hard work of the editor and reviewers!

Yours sincerely,

Xueling Li

Xueling Li, PhD

State Key Laboratory of Reproductive Regulation & Breeding of Grassland Livestock

Research Center for Laboratory Animal Science

Inner Mongolia University

No. 24 Zhao Jun Rd

Yu Quan Qu, Hohhot, Inner Mongolia, P. R. China 010070

Phone: +86 471 3679807

Fax: +86 471 3679807

E-mail: lixueling@imu.edu.cn

Supplement:

Original Immunoblots for Western blotting

PRDM14, P300, G9A, GAPDH for figure 7C

H3K4me1, H3K4me2, H3K27me3 and H3 for figure 7D

DNMT3A, DNMT3B,GAPDH for figure 9B

Cells were collected and protein was extracted by Mammalian Protein Extraction Reagent (CWBIO, China) according to the manufacturer’s procedure. The product is added to the Loading Buffer and boiled in boiling water for 5 minutes for denaturation. Load the same amount of protein sample into SDS-PAGE gel and electrophoresis at 90 - 120 V for 30 min - 3 h. After maker separated, the gel is cut according to the marker position to obtain corresponding protein sizes, and transferred to nitrocellulose membrane, at constant current 200 mA for 30 min-1 h. After the transfer of the required strip is completed, place it in 5% sealing solution and place it at room temperature for 1 h. Transfer the membrane to the diluent of primary antibody at 4℃ and incubated overnight, and then wash the membrane with TBST solution for 3 times. The membrane was incubated with second antibody at room temperature for 1 h, and washed with TBST for 3 times. After treatment the membrane is exposed in the indicator. The anti-bodies used in this experiment include anti-H3 (4620S, Cell Signaling Technology), anti-H3K4me1 (ab8895, Abcam), anti-H3K4me2 (ab32356, Abcam), anti-H3K27me3 (ab6002, Abcam), anti-GAPDH (10494-1-AP, Proteintech), anti-PRDM14 (ab187881, Abcam), anti-P300 (ab54984, Abcam), and anti-G9A (ab18894, Abcam), anti-DNMT3A (D23G1, Cell Signaling Technology), anti-DNMT3B (bs-0301R, Bioss). Images were obtained using a Tanon 5200 Multi Automatic Fluorescence and Chemiluminescence Imaging System (Tanon, China).

this experiment include anti-H3 (4620S, Cell Signaling Technology), anti-H3K4me1 (ab8895, Abcam), anti-H3K4me2 (ab32356, Abcam), anti-H3K27me3 (ab6002, Abcam), anti-GAPDH (10494-1-AP, Proteintech), anti-PRDM14 (ab187881, Abcam), anti-P300 (ab54984, Abcam), and anti-G9A (ab18894, Abcam), anti-DNMT3A (D23G1, Cell Signaling Technology), anti-DNMT3B (bs-0301R, Bioss). Images were obtained using a Tanon 5200 Multi Automatic Fluorescence and Chemiluminescence Imaging System (Tanon, China).

Round 2

Reviewer 2 Report

-

-